# Training Deep Learning Models with Norm-Constrained LMOs

**Thomas Pethick** [1] **Wanyun Xie** [1] **Kimon Antonakopoulos** [1]
**Zhenyu Zhu** [1] **Antonio Silveti-Falls** [2] **Volkan Cevher** [1]

## Abstract

In this work, we study optimization methods that leverage the linear minimization oracle (lmo) over a norm-ball. We propose a new stochastic family of algorithms that uses the lmo to adapt to the geometry of the problem and, perhaps surprisingly, show that they can be applied to unconstrained problems. The resulting update rule unifies several existing optimization methods under a single framework. Furthermore, we propose an explicit choice of norm for deep architectures, which, as a side benefit, leads to the transferability of hyperparameters across model sizes. Experimentally, we demonstrate significant speedups on nanoGPT training using our algorithm, Scion, without any reliance on Adam. The proposed method is memory-efficient, requiring only one set of model weights and one set of gradients, which can be stored in half-precision. The code is available at https://github.com/LIONS-EPFL/scion.

## 1. Introduction

Deep learning has greatly benefited from adaptive optimization methods such as RMSProp (Hinton et al., 2012), AdaGrad (Duchi et al., 2011; McMahan & Streeter, 2010), and Adam (Kingma, 2014), which dynamically change the geometry of the problem based on gradients encountered on-the-fly during training. While these methods have demonstrated remarkable success, they fundamentally treat neural networks (NNs) as optimization problems where we lack any prior knowledge about their particular setting.

However, NNs are far from being black boxes—their structure is not only known but they are deliberately designed. This simple observation raises directly the question:

*Is it more beneficial to adapt the optimizer a priori, instead of exploring their respective geometries on-the-fly?*

[1]LIONS, EPFL [2]CVN, Université Paris-Saclay. Correspondence to: Thomas Pethick <thomas.pethick@epfl.ch>.

*Proceedings of the 42nd International Conference on Machine Learning*, Vancouver, Canada. PMLR 267, 2025. Copyright 2025 by the author(s).

Adaptation on-the-fly has been the defacto standard in this setting, with adaptive algorithms, such as Adam (Kingma, 2014), dominating the deep learning model training.

One possible way for adaptation a priori, which we focus on in this work, is to modify the underlying norm used to measure distances in the parameter space. There is precedence to our proposal, as the early work by Carlson et al. (2015a;b;c) introduced the stochastic spectral descent method (SSD), which performs steepest descent in the spectral norm, and demonstrated that the method can substantially accelerate deep learning training.

The significance of the SSD approach has been very recently brought back to attention by Bernstein & Newhouse (2024b), who showed that the Shampoo optimizer (Gupta et al., 2017)—winner of the external tuning track at the 2024 AlgoPerf: Training Algorithms competition (Dahl et al., 2023)—can be viewed as SSD when a certain accumulation is disabled. Moreover, Bernstein & Newhouse (2024b) introduced an efficient Newton-Schultz iteration to replace the approximate SVD calculations previously required. Jordan et al. (2024b) incorporated the Newton-Schultz iteration with additional momentum into SSD under the name Muon to achieve impressive results on the nanoGPT architecture by applying it to the hidden layers.

**Contributions** This work focuses on developing an algorithmic framework that can exploit an appropriate choice of norm for the entire neural network with particular emphasis on hyperparameter transfer across model sizes (Yang & Hu, 2021), convergence and practical performance.

To adapt to the geometry a priori, we will build on a classical (but unexpected) family of algorithms in contrast to the steepest descent methods, namely the ones involving the linear minimization oracle (lmo) over a norm-ball constraint known as the Conditional Gradient (CG) methods.

While classically being used for constrained problems, we take the slightly unusual approach by showing that the lmos can be used even for unconstrained problems. The algorithm, dubbed as the unconstrained Stochastic Conditional Gradient method (uSCG), shows improvements both theoretically and practically when the norm-ball constraint matches the natural geometry of the problem.

*Table 1.* Special instantiations of uSCG according to different choices of norm. The reduced SVD is given as $d = U \operatorname{diag}(\sigma) V^\top$. Weight decay is captured by SCG, which provides explicit control on the norm of the parameters.

| Method | $\alpha_k$ | Problem | lmo constraint set $\mathcal{D}$ | lmo | Reference |
|---|---|---|---|---|---|
| Normalized SGD | 1 | Unconstrained | Euclidean $\|\cdot\|_2$-ball | $-\rho\frac{d}{\|d\|_2}$ | (Hazan et al., 2015) |
| Momentum Normalized SGD | $[0,1]$ | Unconstrained | Euclidean $\|\cdot\|_2$-ball | $-\rho\frac{d}{\|d\|_2}$ | (Cutkosky & Mehta, 2020) |
| SignSGD | 1 | Unconstrained | Max-norm $\|\cdot\|_\infty$-ball | $-\rho\operatorname{sign}(d)$ | (Bernstein et al., 2018, Thm. 1)[2] |
| Signum | $[0,1]$ | Unconstrained | Max-norm $\|\cdot\|_\infty$-ball | $-\rho\operatorname{sign}(d)$ | (Bernstein et al., 2018, Thm. 3)[2] |
| Muon[1] | $[0,1]$ | Unconstrained | Spectral $\|\cdot\|_{\mathcal{S}_\infty}$-ball | $-\rho UV^\top$ | (Jordan et al., 2024b) |

[1] With non-Nesterov based momentum. [2] The theoretical guarantee relies on increasing batch size.

In particular, we build on the Stochastic Conditional Gradient (SCG) method of Mokhtari et al. (2020) from the constrained setting, which provides explicit control on the norm of NN weight matrices. This is particularly relevant for robust image classification (Cisse et al., 2017), generalization bounds (Bartlett et al., 2017), Lipschitz control of generative adversarial networks (Arjovsky et al., 2017; Miyato et al., 2018), diffusion models (Karras et al., 2024, Sec. 2.3), and for ensuring Lipschitz continuity of NNs (Large et al., 2024).

Concretely, we make the following contributions:

*Theoretical rates:* We introduce a new, stochastic lmo based family of algorithms uSCG, which can exploit the specific geometry of the problem. In doing so we achieve the $O(n^{-1/4})$ order optimal convergence rate under general nonconvexity and stochasticity for uSCG (Arjevani et al., 2022). Moreover, we provide a new analogous guarantee for the constrained case for SCG. A major benefit of both methods is that their stepsize is agnostic to the Lipschitz constant, in contrast to steepest descent which requires the stepsize to be taken small enough.

*Unification*: Our lmo-based approach provides a unifying framework for various popular algorithms, based on the norm choice (see Table 1); as a byproduct we establish the first provable rate for the Muon optimizer with and without weight decay. More importantly, this generality allows us to design a new method for deep learning based on operator norms called SCION (Algorithm 3), which enjoys zero-shot hyperparameter transferability (Yang et al., 2022), and can be implemented storing only one set of parameters and one gradient (stored in half-precision), economizing on memory in large-scale training.

*Numerical validation*: We carry out exhaustive numerical evaluation of SCION ranging from small scale experiments on MLPs and CNNs to ViT on ImageNet and NanoGPT models with up to 3B parameters. We consistently observe the transferability properties across all settings for SCION. The scheme is more tolerant to large batch sizes and exhibits superior performance due to the a priori adaptation.

An additional lmo-based algorithm ALMOND can be found in Appendix D.3, generalizing the Normalized SGD based method of Zhao et al. (2020), for training with large-batches. Key differences of ALMOND with uSCG and SCG are discussed to further motivate our algorithms.

## 2. Preliminaries

We are interested in solving the following general (possibly nonconvex) optimization problem

$$\min_{x \in \mathcal{X}} f(x), \tag{1}$$

where $f$ is smooth in some not necessarily Euclidean norm and the problem is either unconstrained (e.g., $\mathcal{X} = \mathbb{R}^d$) or constrained to $\mathcal{X} = \mathcal{D}$ where $\mathcal{D}$ is the norm-ball defined as

$$\mathcal{D} := \{x \mid \|x\| \le \rho\}.$$

The central primitive in the algorithms considered in this work is the linear minimization oracle (lmo) defined as

$$\operatorname{lmo}(s) \in \arg\min_{x \in \mathcal{D}} \langle s, x \rangle, \tag{2}$$

where we are particularly interested in the special case where the constraint set is a norm constraint $\|x\| \le \rho$, for some $\rho > 0$ and some norm $\|\cdot\|$, which does not have to be the Euclidean norm. Examples of norm-constrained lmos are provided in Table 1 and Table 2 regarding operator norms. An important property of the lmo is that the operator is scale invariant, i.e., $\operatorname{lmo}(a \cdot s) = \operatorname{lmo}(s)$ for $a > 0$, and in fact we have by construction under the norm constraints that $\|\operatorname{lmo}(s)\| \le \rho$. Thus, it is only the direction of the input $s$ that matters and not the magnitude.

A classical method for solving the constrained variant of problem 1, when the lmo is available, is the Conditional Gradient method (CG) (Frank et al., 1956; Clarkson, 2010; Jaggi, 2013), which proceeds as follows with $\gamma_k \in (0,1)$

$$x^{k+1} = (1 - \gamma_k)x^k + \gamma_k \operatorname{lmo}(\nabla f(x^k)), \tag{CG}$$

ensuring the feasibility of $x^k$ via simplicial combination.

---

**Algorithm 1** Unconstrained SCG (uSCG)

---

**Input:** Horizon $n$, initialization $x^1 \in \mathcal{X}$, $d^0 = 0$, momentum $\alpha_k \in (0, 1]$, and stepsize $\gamma_k \in (0, 1)$

1: **for** $k = 1, \ldots, n$ **do**
2:     Sample $\xi_k \sim \mathcal{P}$
3:     $d^k \leftarrow \alpha_k \nabla f(x^k, \xi_k) + (1 - \alpha_k)d^{k-1}$
4:     $x^{k+1} \leftarrow x^k + \gamma_k \operatorname{lmo}(d^k)$
5: Choose $\bar{x}^n$ uniformly at random from $\{x^1, \ldots, x^n\}$
**Return** $\bar{x}^n$

---

**Algorithm 2** Stochastic Conditional Gradient (SCG)

---

**Input:** Horizon $n$, initialization $x^1 \in \mathcal{D}$, $d^0 = 0$, momentum $\alpha_k \in (0, 1]$, and stepsize $\gamma_k \in (0, 1)$

1: **for** $k = 1, \ldots, n$ **do**
2:     Sample $\xi_k \sim \mathcal{P}$
3:     $d^k \leftarrow \alpha_k \nabla f(x^k, \xi_k) + (1 - \alpha_k)d^{k-1}$
4:     $x^{k+1} \leftarrow (1 - \gamma_k)x^k + \gamma_k \operatorname{lmo}(d^k)$
5: Choose $\bar{x}^n$ uniformly at random from $\{x^1, \ldots, x^n\}$
**Return** $\bar{x}^n$

---

Usually, the CG is attractive when the constraint set is an atomic set (e.g., the $\ell_1$-norm ball) in which case each update may be efficiently stored. Our focus lies in the more unconventional cases of the vector $\ell_\infty$-norm ball and spectral norm ball for which the updates are in contrast dense. Furthermore, we are interested in the unconstrained case in addition to the constrained problem which CG solves.

In the stochastic regime, the analyzing lmo-based algorithms is involved. Even when the stochastic oracle $\nabla f(x, \xi)$ is unbiased, the direction of the updates, as defined by $\operatorname{lmo}(\nabla f(x, \xi))$, is not unbiased. To help overcome this difficulty, we will employ a commonly used trick of averaging past gradients with $\alpha_k \in (0, 1]$ (aka momentum),

$$d^k = (1 - \alpha_k)d^{k-1} + \alpha_k \nabla f(x^k, \xi_k), \qquad (3)$$

which will rigorously help with algorithmic convergence.

## 3. Our Methods

For the unconstrained case we introduce a new method, dubbed the unconstrained SCG method (uSCG):

$$x^{k+1} = x^k + \gamma_k \operatorname{lmo}(d^k) \qquad \text{(uSCG)}$$

with stepsizes $\gamma_k \in (0, 1)$. Instead of the convex combination in CG, the update rule simply sums the lmos. In contrast with e.g., gradient descent, the update always has the same magnitude regardless of the size of the gradient average $d^k$. The final algorithm is presented in Algorithm 1.

For the constrained case, we revisit the SCG method of Mokhtari et al. (2020) and adopt it for the non-convex objectives typically encountered in deep learning model training. This algorithm (Algorithm 2) proceeds as follows

$$x^{k+1} = (1 - \gamma_k)x^k + \gamma_k \operatorname{lmo}(d^k) \qquad \text{(SCG)}$$

with stepsizes $\gamma_k \in (0, 1)$.

**Connection to weight decay** For uSCG, weight decay has a very precise interpretation, since the method reduces to SCG. Consider the following variant of uSCG with weight decay $x^{k+1} = x^k + \gamma_k \operatorname{lmo}(d^k) - \gamma_k \mu x^k$.

The weight decay parameter $\mu \in [0, 1]$ interpolates between uSCG and SCG. If the weight decay is in $(0, 1)$ then the algorithm is still an instance of SCG and thus solve a constrained problem, but one with a larger radius of $\rho' = \frac{\rho}{\mu}$ with a stepsize chosen as $\gamma'_k = \gamma_k \mu$.

Therefore, all schemes in Table 1 guarantees a norm bound of $\frac{\rho}{\mu}$ on the parameters when combined with weight decay. The connection between weight decay and constrained optimization, in the special case where $\operatorname{lmo} = \operatorname{sign}$ (when the norm-constraint in (2) is the vector $\ell_\infty$-norm) has also been observed in Xie & Li (2024); D'Angelo et al. (2023). Due to the fixed magnitude of the lmo both methods provides a guarantee on the maximum norm of the parameters.

> **Insight 3.1.** Both uSCG and SCG provide explicit control on the norm of the parameters:
>
> (i) SCG guarantees $\|x\| \leq \rho$.
>
> (ii) uSCG guarantees $\|x\| \leq \rho \sum_{k=1}^n \gamma_k$.

Norm control is particularly useful for long runs (*cf.* Figure 9) and to avoid overfitting in multi-epoch training (*cf.* Figures 10 and 11 regarding CIFAR10 experiments).

### 3.1. Choice of Norm Constraint

To choose an appropriate norm for deep learning, we build on the operator norm perspective of Large et al. (2024); Bernstein & Newhouse (2024a). To simplify the presentation we will consider a linear MLP as a running example, but in Section 3.2, we point to our theoretical guarantees with activation functions.

Let us a consider a linear MLP with the initial hidden layer defined as $h_1(z) = W_1 z + b_1$ and the remaining layers

$$h_\ell(z) = W_\ell h_{\ell-1}(z) + b_\ell, \quad \forall \ell \in 2, .., L;$$

with $b_L = 0$. We denote the global loss as $\mathcal{L}(h_L(z), y)$ where $\mathcal{L}$ is the loss function and $y$ is a 1-hot encoded target vector. We use the overloaded notation $W_\ell \in \mathbb{R}^{d_{\text{out}} \times d_{\text{in}}}$, where $d_{\text{out}}$ and $d_{\text{in}}$ implicitly have dependency on $\ell$ and can thus be distinct across different layers.

*Table 2.* Example operator norms and the associated lmos of a matrix $A \in \mathbb{R}^{d_{\text{out}} \times d_{\text{in}}}$. The reduced SVD is given as $A = U \operatorname{diag}(\sigma) V^\top$, sign acts elementwise, $\operatorname{col}_j(A) := A_{\cdot,j}$ and $\operatorname{row}_i(A) := A_{i,\cdot}$. Note that this table is not exhaustive.

| | $1 \to$ RMS (ColNorm) | $1 \to \infty$ (Sign) | RMS $\to$ RMS (Spectral) | RMS $\to \infty$ (RowNorm) |
|---|---|---|---|---|
| **Norm** | $\max_j \frac{1}{\sqrt{d_{\text{out}}}} \| \operatorname{col}_j(A) \|_2$ | $\max_{i,j} |A_{i,j}|$ | $\sqrt{d_{\text{in}}/d_{\text{out}}} \|A\|_{\mathcal{S}_\infty}$ | $\max_i \sqrt{d_{\text{in}}} \| \operatorname{row}_i(A) \|_2$ |
| **LMO** | $\operatorname{col}_j(A) \mapsto -\sqrt{d_{\text{out}}} \frac{\operatorname{col}_j(A)}{\| \operatorname{col}_j(A) \|_2}$ | $A \mapsto -\operatorname{sign}(A)$ | $A \mapsto -\sqrt{d_{\text{out}}/d_{\text{in}}} UV^\top$ | $\operatorname{row}_i(A) \mapsto -\frac{1}{\sqrt{d_{\text{in}}}} \frac{\operatorname{row}_i(A)}{\| \operatorname{row}_i(A) \|_2}$ |

*Table 3.* The choice of lmo can be different between layers and can depend on the assumptions on the input. For simplicity we overload notation and write the reduced SVD as $W_\ell = U \operatorname{diag}(\sigma) V^\top \in \mathbb{R}^{d_{\text{out}} \times d_{\text{in}}}$ for all $\ell \in [L]$.

| Parameter | $W_1$ (image domain) | $\{W_\ell\}_{\ell \in [2,...,L-1]}$ | | $W_L$ | | $b_\ell$ |
|---|---|---|---|---|---|---|
| **Norm** | RMS $\to$ RMS | RMS $\to$ RMS | RMS $\to$ RMS | RMS $\to \infty$ | $1 \to \infty$ | RMS |
| **LMO** | $-\max(1, \sqrt{d_{\text{out}}/d_{\text{in}}}) UV^\top$ | $-\sqrt{d_{\text{out}}/d_{\text{in}}} UV^\top$ | $-\sqrt{d_{\text{out}}/d_{\text{in}}} UV^\top$ | $\operatorname{row}_i(W_L) \mapsto -\frac{1}{\sqrt{d_{\text{in}}}} \frac{\operatorname{row}_i(W_L)}{\| \operatorname{row}_i(W_L) \|_2}$ | $-\frac{1}{d_{\text{in}}} \operatorname{sign}(W_L)$ | $-\frac{b_\ell}{\|b_\ell\|_{\text{RMS}}}$ |
| **Init.** | Semi-orthogonal | Semi-orthogonal | Semi-orthogonal | Row-wise normalized Gaussian | Random sign | 0 |

*Table 4.* Example lmo choices for 1-hot encoded inputs.

| Parameter | $W_1$ (1-hot encoded input) | | |
|---|---|---|---|
| **Norm** | $2 \to$ RMS | $1 \to$ RMS | $1 \to \infty$ |
| **LMO** | $-\sqrt{d_{\text{out}}} UV^\top$ | $\operatorname{col}_j(W_1) \mapsto -\sqrt{d_{\text{out}}} \frac{\operatorname{col}_j(W_1)}{\| \operatorname{col}_j(W_1) \|_2}$ | $-\operatorname{sign}(W_1)$ |
| **Init.** | Semi-orthogonal | Column-wise normalized Gaussian | Random sign |

We need that none of the intermediary hidden states $h_\ell(z)$ blows up by requiring one of the following norm bounds:

(i) $\frac{1}{d_{\text{out}}} \|h_\ell(z)\|_1 \leq 1$      (the average entry is bounded)

(ii) $\|h_\ell(z)\|_{\text{RMS}} \leq 1$      (the typical entry is bounded)

(iii) $\|h_\ell(z)\|_\infty \leq 1$      (the maximum entry is bounded)

where $\|z\|_{\text{RMS}} := \frac{1}{\sqrt{d}} \|z\|_2$ for $z \in \mathbb{R}^d$. Assuming the input to any given layer is bounded in some norm $\|\cdot\|_\alpha$, this requirement corresponds to placing an operator norm constraint on the weight matrices $\{W_\ell\}_{\ell \in [L]}$ and a norm constraint on the biases $\{b_\ell\}_{\ell \in [L-1]}$.

The operator norm is in turn defined as follows

$$\|A\|_{\alpha \to \beta} := \max_{z \in \mathbb{R}^d, z \neq 0} \frac{\|Az\|_\beta}{\|z\|_\alpha} = \sup_{\|z\|_\alpha = 1} \|Az\|_\beta. \quad (4)$$

Directly from the definition, we have that if the input $z$ is bounded through $\|z\|_\alpha \leq 1$, then the output $\|Az\|_\beta$ will be bounded when $\|A\|_{\alpha \to \beta}$ is bounded.

A collection of operator norms and their resulting lmos is provided in Table 2. It will be convenient to convert between bounds on these different operator norm which the following fact makes precise.

**Fact 3.1.** *The operator norm satisfies for some $\rho > 0$*

(i) $\|z\|_\beta \leq \rho \|z\|_c, \forall z \in \mathbb{R}^d \Rightarrow \|A\|_{\alpha \to \beta} \leq \rho \|A\|_{\alpha \to c}.$

(ii) $\|z\|_\alpha \geq \frac{1}{\rho} \|z\|_c, \forall z \in \mathbb{R}^d \Rightarrow \|A\|_{\alpha \to \beta} \leq \rho \|A\|_{c \to \beta}.$

Fact 3.1 tells us that we can bound operator norms using bounds on the vector norms, i.e.,

$$\|z\|_\infty \leq \|z\|_2 \leq \|z\|_1 \leq \sqrt{d}\|z\|_2 \leq d\|z\|_\infty, \quad \forall z \in \mathbb{R}^d. \quad (5)$$

We start by focusing on controlling the RMS norm, but will later consider other norms. There are three example operator norms to consider for the MLP in consideration:

(i) Initial layer $h_1(z)$: $\|W_1\|_{\alpha_1 \to \text{RMS}} \leq 1$.

(ii) Intermediary layers $h_\ell(z)$: $\|W_\ell\|_{\text{RMS} \to \text{RMS}} \leq 1$ $\forall \ell \in \{2, .., L-1\}$.

(iii) Last layer $h_L(z)$: $\|W_L\|_{\text{RMS} \to \beta_L} \leq 1$.

Note that the operator norm $\|\cdot\|_{\text{RMS} \to \text{RMS}}$ is a scaled spectral norm, i.e., $\|A\|_{\text{RMS} \to \text{RMS}} = \sqrt{d_{\text{in}}/d_{\text{out}}}\|A\|_{2 \to 2} = \sqrt{d_{\text{in}}/d_{\text{out}}}\|A\|_{\mathcal{S}_\infty}$ for $A \in \mathbb{R}^{d_{\text{out}} \times d_{\text{in}}}$.

To concisely write the layerwise norm constraints in terms of a norm constraint on the joint parameter $x = \{W_\ell, b_\ell\}_{\ell \in [L]}$, we can define the norm in the lmo (2) as

$$\|x\| := \max_{\ell \in [L]} \frac{1}{\rho_\ell} \max\{\|W_\ell\|_{\alpha_\ell \to \beta_\ell}, \|b_\ell\|_{\beta_\ell}\} \leq 1 \quad (6)$$

where $\rho_\ell$ is a layerwise scaling factor of the constraint radius. What is particularly convenient algorithmically for the $\ell_\infty$-norm is that the layerwise lmos can be computed separately (cf. Algorithm 3). A norm choice across layers was made through the $\ell_1$-norm in Flynn (2017) and through the $\ell_\infty$-norm with the *modular norm* (Large et al., 2024) and block-normalization (Balles et al., 2020; Yu et al., 2017; Ginsburg et al., 2019).

A choice needs to be made for the input norm $\alpha_1$ and output norm $\beta_L$, which depends on the application:

**Input layer** For image domains, usually the input is rescaled pixel-wise to e.g., ensure that $z \in [-1, 1]$ in which

case $\|z\|_{\mathrm{RMS}} \leq 1$ and the appropriate operator norm for the first layer becomes $\|W_1\|_{\mathrm{RMS}\to\mathrm{RMS}} = \sqrt{d_{\mathrm{in}}/d_{\mathrm{out}}}\|W_1\|_{\mathcal{S}_\infty}$. In order to deal with the case where $d_{\mathrm{in}} > d_{\mathrm{out}}$, we choose the radius to be $\max(1, \sqrt{d_{\mathrm{out}}/d_{\mathrm{in}}})$ (*cf.*, Appendix B.1).

For language tasks, the input $z$ is usually a 1-hot encoded vector in which case $\|z\|_\infty = \|z\|_2 = \|z\|_1 = 1$. In turn, $\|W_1\|_{\infty\to\mathrm{RMS}} = \|W_1\|_{2\to\mathrm{RMS}} = \|W_1\|_{1\to\mathrm{RMS}}$ holds on this restricted domain, where we can freely pick the operator norm that leads to the simplest update rule (Table 2).

The simplest form for the lmo is arguably induced by $\|\cdot\|_{1\to\mathrm{RMS}}$ since the lmo can be computed exactly, while from $\|\cdot\|_{2\to\mathrm{RMS}}$ we can observe a more aggressive scaling factor in the lmo, $-\sqrt{d_{\mathrm{out}}}UV^\top$, than the $-\sqrt{d_{\mathrm{out}}/d_{\mathrm{in}}}UV^\top$ used in intermediate layers. The norm choice $\|\cdot\|_{1\to\mathrm{RMS}}$ was first proposed in Large et al. (2024) for 1-hot encoded input. Through the above reasoning we see how the norm is equivalent to an appropriately scaled spectral norm.

**Output layer**  For the final layer, we are not restricted to bounding the output in $\ell_{\mathrm{RMS}}$ and can alternatively choose bounding the maximal entry through $\ell_\infty$. Additionally, we can bound $\|A\|_{\mathrm{RMS}\to\infty} \leq \frac{1}{d_{\mathrm{in}}}\|A\|_{1\to\infty}$, by using (5) through Fact 3.1, which leads to a dimension scaled sign update rule for the last layer.

We summarize the different norm choices and their resulting lmos in Tables 3 and 4. Table 3 provides an overview of norm choices of output layers, while Table 4 provides choices for input layers under 1-hot encoded input.

Provided that the input is bounded as described, each of the hidden states $h_\ell(z)$ and logits $h_L(z)$ will be bounded in the RMS norm. In order to ensure feasibility of the initialization in the constrained case when employing SCG, we propose to initialize on the boundary similar to Large et al. (2024) (see Appendix B for details).

**Insight 3.2**.

(i) For 1-hot encoded input, ColNorm and Spectral are equivalent for the first layer, in which case ColNorm is favored since the lmo can be computed exactly.

(ii) Sign can be used both for the first and last layer which is crucial for weight sharing.

(iii) To transfer learning rate from proxy models when the width is smaller than the input dimension it is important to rescale the lmo as $\max(1, \sqrt{d_{\mathrm{out}}/d_{\mathrm{in}}})$.

These observations leads to the recommendations below.

**Recommendation 3.1**. We refer to the instantiation of uSCG and SCG using operator norms as UNCONSTRAINED SCION and SCION respectively (*cf.* Algorithm 3), which stands for **S**tochastic **C**onditional Grad**i**ent with **O**perator **N**orms. We recommend the following configurations of the layer norms (First layer → Intermediary layers → Last layer):

(i) image domains: Spectral → Spectral → Sign

(ii) 1-hot input:  ColNorm → Spectral → Sign

(iii) weight sharing:  Sign → Spectral → Sign

The lmo names are defined in Table 2 and weight sharing refers to parameter sharing between the first and last layer. Each layer should be scaled appropriately according to Tables 3 and 4.

**Other norm choices**  So far our argument has been based on the invariance provided by $\|\cdot\|_{\mathrm{RMS}\to\mathrm{RMS}}$ for intermediary layers: i.e., the RMS norm of the output of layer $\ell$ is bounded, so the input of next layer $\ell+1$ is also bounded in the RMS norm. Since the lmo of $\|\cdot\|_{\mathrm{RMS}\to\mathrm{RMS}}$ can be computed efficiently we can directly use this norm choice for our update rule. However, it is possible to choose another norm such as $\|\cdot\|_{1\to\mathrm{RMS}}$, as long as the RMS norm guarantee on the output of $W_\ell$ is converted into a guarantee on the $\ell_1$-norm of the input of layer $\ell+1$. Specifically, we have that $\|\cdot\|_{\mathrm{RMS}\to\mathrm{RMS}} \leq d_{\mathrm{in}}\|\cdot\|_{1\to\mathrm{RMS}}$ through Fact 3.1. Alternatively, we can rely on the invariance provided by $\|\cdot\|_{\infty\to\infty}$, for which Fact 3.1 tells us that $\|\cdot\|_{\infty\to\infty} \leq \sqrt{d_{\mathrm{in}}}\|\cdot\|_{\mathrm{RMS}\to\infty}$ and $\|\cdot\|_{\infty\to\infty} \leq d_{\mathrm{in}}\|\cdot\|_{1\to\infty}$. We obtain methods exclusively relying on RowNorm, ColNorm and Sign as summarized in Table 6 of Appendix B.

### 3.2. Hyperparameter Transfer

The intuition behind why (UNCONSTRAINED) SCION may enjoy hyperparameter transfer is suggested by the spectral scaling rule of Yang et al. (2023), which states that feature learning may be ensured by requiring that, for MLPs with weight matrices $W_\ell \in \mathbb{R}^{d_{\ell-1} \times d_\ell}$, the following holds:

$$\|W_\ell\|_{\mathcal{S}_\infty} = \Theta\left(\sqrt{\tfrac{d_\ell}{d_{\ell-1}}}\right) \text{ and } \|\Delta W_\ell\|_{\mathcal{S}_\infty} = \Theta\left(\sqrt{\tfrac{d_\ell}{d_{\ell-1}}}\right)$$

where $\|\cdot\|_{\mathcal{S}_\infty}$ denotes the spectral norm and $\Delta W_\ell$ is the update change. For uSCG, the update change is given by $x^{k+1} - x^k = \gamma\,\mathrm{lmo}(d^k)$, so the requirement is automatically satisfied by the spectral norm choice from Table 3.

We formalize this intuition in Lemma C.6 of Appendix C.2 following the proof technique of Yang et al. (2023), which holds for losses including logistic regression and MSE,

and activation functions including ReLU, GELU and Tanh. Specifically, we show that the so-called maximal update learning rate $\gamma^*$ (i.e., the learning rate that enables the hidden layer preactivations to undergo the largest possible change in a single update step) is independent of width. Our theoretical analysis is conducted under a simplified setting with momentum $\alpha_k = 1$ in Algorithm 1, which we adopt to facilitate a clean derivation of the width-invariant property. Thus, a learning rate tuned on a smaller model can be directly applied to a wider model without compromising the maximal update property.

## 4. Related Works

**Hyperparameter transfer** Yang & Hu (2021); Yang et al. (2022) showed that there exists a parameterization (i.e., a choice of initialization and layerwise stepsize scaling) for which the features in every single layer evolve in a width-independent manner. The so-called Maximal Update Parametrization ($\mu$P) allows transferring optimal hyperparameter from a small proxy model to a large model.

A relationship with the spectral norm was established in Yang et al. (2023). An operator norm perspective was taken in the modular norm framework of Large et al. (2024); Bernstein & Newhouse (2024a), which was used to show Lipschitz continuity with constants independent of width. We build on this perspective and propose the $1 \to \infty$ operator norm and RMS $\to \infty$, which leads to a sign update rule and row normalization respectively.

**Steepest descent in a normed space** Steepest descent in a possibly non-Euclidean space can be written in terms of the lmo (*cf.*, Appendix A.1) provided a stream of stochastic gradients $(g^k)_{k\in\mathbb{N}}$ and an initialization $x^0 \in \mathcal{X}$,

$$x^{k+1} = x^k - \gamma[g^k]^\sharp = x^k + \frac{\gamma}{\rho}\|g^k\|_* \operatorname{lmo}(g^k), \quad (7)$$

where $[\cdot]^\sharp := \arg\max_{x\in\mathcal{X}} \langle\cdot, x\rangle - \frac{1}{2}\|x\|^2$ is the sharp-operator (Nesterov, 2012; Kelner et al., 2014).

The deterministic case is analyzed in Nesterov (2012); Kelner et al. (2014), and was extended to the stochastic case in Carlson et al. (2015b), with a particular empirically focus on the spectral norm, named as (pre-conditioned) stochastic spectral descent (SSD) (Carlson et al., 2015a;b;c). Their SSD algorithm is an instance of the Majorization-Minimization (MM) algorithms (Lange, 2016), which iteratively minimizes a locally tight upper bound. The dualization in Bernstein & Newhouse (2024a) is also motivated by the sharp-operator.

In contrast to (7), uSCG and SCG are invariant to the magnitude of the gradients and do not need to compute the dual norm $\|\cdot\|_*$, which cannot be computed independently across layers. Intuitively, the scale invariance of the lmo allows convergence to be established without knowledge of the Lipschitz constant $L$ (*cf.* Theorems 5.4 to 5.7).

Unlike the lmo-based schemes, extending sharp-operator-based algorithms to handle constrained problems is non-trivial even in the vector case (El Halabi, 2018). Additionally, a practical concern of using specifically spectral norm projections in deep learning is that the model weights themselves can be dense (so the required SVD would be expensive), while gradients used in the lmo are usually low-rank (allowing efficient SVD approximations).

**Muon** The Muon optimizer (Jordan et al., 2024b) is introduced as a steepest descent method. The implementation interestingly ignores the scaling $\|\cdot\|_*$ appearing in the update (*cf.*, (7)), so Muon is effectively using the lmo over the spectral norm instead of the sharp operator. Provided a stream of stochastic gradients $(g^k)_{k\in\mathbb{N}}$ and an initialization $x^0 \in \mathcal{X}$ the Muon optimizer can then be written as follows

$$G^k = g^k + \beta G^{k-1} \quad \text{(Muon)}$$
$$x^{k+1} = \begin{cases} x^k + \gamma\operatorname{lmo}(g^k + \beta G^k) & \text{if Nesterov} \\ x^k + \gamma\operatorname{lmo}(G^k) & \text{otherwise} \end{cases}$$

where the lmo corresponds implicitly to the spectral norm.

The accumulation $G^k$ can be written in terms of the averaged gradient $d^k$ in Algorithm 2. We have that $d^k = \alpha G^k$ by picking $\alpha = (1 - \beta)$. Since the lmo is scale invariant, $d^k$ and $G^k$ can be used interchangeably without changing the update. Thus, we can alternatively write Muon (with non-Nesterov based momentum) exactly as uSCG.

In practice Muon is only applied to hidden layers, thus excluding the first layer and the last layer for which Adam(W) or SGD is used. In contrast, we apply uSCG and SCG to all layers and demonstrate transferability of the stepsize.

**Moonlight** Since the first version of this paper appeared on arXiv on the 11th of February, Liu et al. (2025) also proposed integrating Muon with weight decay to control the norm of the parameters. They use the same scaling factor of the lmo, $\sqrt{\max\{d_{\text{in}}, d_{\text{out}}\}}$, as the Muon baseline we compare against (*cf.* Appendix E.4).

**MARS** The MARS-Shampoo optimizer (Yuan et al., 2024, Alg. 4) can be seen as an instance of SCG with spectral norm constraints, but using the STORM gradient estimator (Cutkosky & Orabona, 2019) instead of (3).

**Sign** SignSGD and the momentum variant Signum were brought to prominence and further analyzed in Bernstein et al. (2018) motivated by efficient communication for distributed optimization, while they are originally introduced with the dual norm scaling and used only for weight bias updates in Carlson et al. (2015a;b;c). These schemes are typically studied under the framework of steepest descent,

which results in the $\|g^k\|_1$ stepsize scaling in (7) usually not present in practice as remarked in Balles et al. (2020).

**Normalization**   The LARS optimizer (You et al., 2017) uses normalized gradient and was shown to be particularly useful for large batch settings. The method can be viewed as performing normalized SGD with momentum (Cutkosky & Mehta, 2020) *layerwise* with a particular adaptive parameter-dependent stepsize.

The layerwise normalization can be captured by uSCG with the norm choice $\max_\ell \|W_\ell\|_F$. The LAMB optimizer (You et al., 2019) incorporates the update into an Adam-like structure. Zhao et al. (2020) considers averaging the normalized gradients rather than the gradients prior to normalization. The update can be written in terms of an lmo, with the (flattened) norm choice $\|x\|_2$, which we generalize with a new algorithm in Appendix D.3 to arbitrary norms.

**Continuous greedy**   With zero initialization, $x_1 = 0$, and stepsize $\gamma_k = \gamma = 1/n$, uSCG recovers the stochastic continuous greedy method (Mokhtari et al., 2020; Vondrák, 2008), which can be used to solve DR-submodular maximization problems under Matroid polytope constraints.

**LMO for deep learning**   SCG for training neural networks has been suggested in Pokutta et al. (2020) and Lu et al. (2022), where optimization was specifically constrained to the $K$-sparse polytope with increasing batch-size for handling stochasticity. Beyond these works, we provide convergence guarantees for SCG with constant batch-sizes and, introduce a principled framework for selecting effective constraints based on the input and output space geometries of the layers of the network.

The perturbation in the sharpness-aware minimization (SAM) has been interpreted as an lmo and generalized to arbitrary norms (Pethick et al., 2025), focusing on the max-norm over nuclear norms, $\max_\ell \|W_\ell\|_{\mathcal{S}_1}$.

Scion can also be seen as an instantiation of the Lion-$\mathcal{K}$ (Chen et al., 2023) algorithm with $\partial\mathcal{K}$ chosen to be the lmo. In this case, however, $\mathcal{K}$ is not smooth and the continuous-time analysis presented in Chen et al. (2023) and related works do not apply.

**Trust-region**   The SCG method can be seen as a trust-region method with a linear surrogate. Usually, the surrogate is taken to be quadratic (*cf.* Wright (2006, Ch. 4)). We refer to Conn et al. (2000) for an extensive overview of trust-region methods.

**Preconditioned SGD**   We also recognize the spectral lmo used in SCION as being related to the Preconditioned SGD (PSGD) family of algorithms (Li, 2017; Pooladzandi & Li, 2024) in the sense of whitening the update. In contrast to those methods, we do not keep track of an explicit preconditioner; we compute the lmo at each iteration instead.

**Natural gradient**   Early work on non-Euclidean methods considered measuring the "distance" between models through the Kullback–Leibler (KL) divergence between the output distributions of the models (Amari, 1998). Approximating the KL divergence through a Taylor expansion leads to a steepest descent method, which preconditions with the Fisher information matrix, known as the natural gradient method. Trust-region variants of the natural gradient method were considered with TRPO (Schulman et al., 2015) similarly to how uSCG can be seen as a trust-region variant of steepest descent.

## 5. Analysis

We begin by presenting the two main assumptions we will make to analyze Algorithms 1 and 2. The first is an assumption on the Lipschitz-continuity of $\nabla f$ with respect to the norm $\|\cdot\|_*$ restricted to $\mathcal{X}$. We do not assume this norm to be Euclidean which means our results apply to the geometries relevant to training neural networks.

**Assumption 5.1.** *The gradient $\nabla f$ is L-Lipschitz with $L \in (0, \infty)$, i.e.,*

$$\|\nabla f(x) - \nabla f(x)\|_* \le L\|x - y\| \quad \forall x, y \in \mathcal{X}. \quad (8)$$

*Furthermore, $f$ is bounded below by $f^\star$.*

*Remark* 5.2. Strictly speaking, uSCG only needs Lipschitz continuity to hold locally within a radius $\gamma\rho$, since the assumption is only invoked between two consecutive iterates.

Our second assumption is that the stochastic gradient oracle we have access to is unbiased and has a bounded variance, a typical assumption in stochastic optimization.

**Assumption 5.3.** *The stochastic gradient oracle $\nabla f(\cdot, \xi) : \mathcal{X} \to \mathbb{R}^d$ satisfies.*

*(i) Unbiased:* $\mathbb{E}_\xi [\nabla f(x, \xi)] = \nabla f(x) \quad \forall x \in \mathcal{X}$.

*(ii) Bounded variance:*
$\mathbb{E}_\xi \left[ \|\nabla f(x, \xi) - \nabla f(x)\|_2^2 \right] \le \sigma^2 \quad \forall x \in \mathcal{X}, \sigma \ge 0$.

With these assumptions, we can state our worst-case convergence rates, first for Algorithm 1 and then for Algorithm 2.

To bridge the gap between theory and practice, we investigate these algorithms when run with a *constant* stepsize $\gamma$, which depends on the specified horizon $n \in \mathbb{N}^*$, and momentum which is either constant $\alpha \in (0, 1)$ (except for the first iteration where we take $\alpha = 1$ by convention) or *vanishing* $\alpha_k \searrow 0$. All results can be extended to *any time* guarantees in a straightforward manner by choosing $\gamma_k$ as

a function of the iteration counter $k$ instead of horizon $n$ and modifying the proofs accordingly.

The exact constants for the rates can be found in the proofs in Appendix D; we try to highlight the dependence on the parameters $L$ and $\rho$, which correspond to the natural geometry of $f$ and $\mathcal{D}$, explicitly here. Our rates are non-asymptotic and use big O notation for brevity.

**Theorem 5.4** (Convergence rate for uSCG with constant $\alpha$). *Suppose Assumptions 5.1 and 5.3 hold. Let $n \in \mathbb{N}^*$ and consider the iterates $\{x^k\}_{k=1}^n$ generated by Algorithm 1 with constant stepsize $\gamma = \frac{1}{\sqrt{n}}$ and constant momentum $\alpha \in (0, 1)$. Then, it holds that*

$$\mathbb{E}[\|\nabla f(\bar{x}^n)\|_*] \leq O\left(\frac{L\rho}{\sqrt{n}} + \sigma\right).$$

**Theorem 5.5** (Convergence rate for uSCG with vanishing $\alpha_k$). *Suppose that Assumptions 5.1 and 5.3 hold. Let $n \in \mathbb{N}^*$ and consider the iterates $\{x^k\}_{k=1}^n$ generated by Algorithm 1 with a constant stepsize $\gamma$ satisfying $\frac{1}{2n^{3/4}} < \gamma < \frac{1}{n^{3/4}}$ and vanishing momentum $\alpha_k = \frac{1}{\sqrt{k}}$. Then, it holds that*

$$\mathbb{E}[\|\nabla f(\bar{x}^n)\|_*] = O\left(\frac{1}{n^{1/4}} + \frac{L\rho}{n^{3/4}}\right).$$

> **Insight 5.1.** The uSCG algorithm remarkably does not require knowledge of the Lipschitz constant $L$, which is intuitively explained by viewing the method as a normalized version of steepest descent through the relationship $\mathrm{lmo}(\cdot) = -\frac{[\cdot]^\sharp}{\|\cdot\|_*}$. Normalizing gradients with the dual norm has also been used in the online learning community to adapt to (local) Hölder smoothness as a simple alternative to AdaGrad-Norm (Orabona, 2023).

These results show that, in the worst-case, running Algorithm 1 with constant momentum $\alpha$ guarantees faster convergence but to a noise-dominated region with radius proportional to $\sigma$. In contrast, running Algorithm 1 with vanishing momentum $\alpha_k$ is guaranteed to make the expected dual norm of the gradient small but at a slower rate. Algorithm 2 exhibits the analogous behavior, as we show next.

Before stating the results for Algorithm 2, we emphasize that they are with *constant* stepsize $\gamma$, which is atypical for conditional gradient methods. However, like most conditional gradient methods, we provide a convergence rate on the so-called Frank-Wolfe gap which measures criticality for the constrained optimization problem over $\mathcal{D}$.

Finally, we remind the reader that the iterates of Algorithm 2 are always feasible for the set $\mathcal{D}$ by the design of the update and convexity of the norm ball $\mathcal{D}$.

**Theorem 5.6** (Convergence rate for SCG with constant $\alpha$). *Suppose Assumptions 5.1 and 5.3 hold. Let $n \in \mathbb{N}^*$ and consider the iterates $\{x^k\}_{k=1}^n$ generated by Algorithm 2 with constant stepsize $\gamma = \frac{1}{\sqrt{n}}$ and constant momentum $\alpha \in (0, 1)$. Then, for all $u \in \mathcal{D}$, it holds that*

$$\mathbb{E}[\langle \nabla f(\bar{x}^n), \bar{x}^n - u \rangle] = O\left(\frac{L\rho^2}{\sqrt{n}} + \sigma\right).$$

**Theorem 5.7** (Convergence rate for SCG with vanishing $\alpha_k$). *Suppose Assumptions 5.1 and 5.3 hold. Let $n \in \mathbb{N}^*$ and consider the iterates $\{x^k\}_{k=1}^n$ generated by Algorithm 2 with a constant stepsize $\gamma$ satisfying $\frac{1}{2n^{3/4}} < \gamma < \frac{1}{n^{3/4}}$ and vanishing momentum $\alpha_k = \frac{1}{\sqrt{k}}$. Then, for all $u \in \mathcal{D}$, it holds that*

$$\mathbb{E}[\langle \nabla f(\bar{x}^n), \bar{x}^n - u \rangle] = O\left(\frac{1}{n^{1/4}} + \frac{L\rho^2}{n^{3/4}}\right).$$

> **Insight 5.2.** For both algorithms, our worst-case analyses for constant momentum suggest that tuning $\alpha$ requires balancing two effects. Making $\alpha$ smaller helps eliminate a constant term that is proportional to the noise level $\sigma$. However, if $\alpha$ becomes too small, it amplifies an $O(1/\sqrt{n})$ term and an $O(\sigma/n)$ term. The stepsize $\gamma$ must also align with the choice of momentum $\alpha$; for vanishing $\alpha_k$ the theory suggests a smaller constant stepsize like $\gamma = \frac{3}{4(n^{3/4})}$ to ensure convergence.

## 6. Experiments

For computing the lmo of layers using a spectral norm constraint, we use the efficient implementation provided in Jordan et al. (2024b) of the Newton-Schultz iteration proposed in Bernstein & Newhouse (2024b). In this section, Muon (Jordan et al., 2024b) refers to the version used in practice, which uses AdamW for the first layer and last layer and Nesterov type momentum.

**GPT** We build on the excellent modded-nanogpt codebase (Jordan et al., 2024a), which makes the following modernizations to Karpathy (2023): rotary embeddings is used instead of positional embeddings, RMS norm is used instead of LayerNorm, and linear decay schedule instead of a cosine stepsize, and the ReLU$^2$ instead of GELU activation function (scaled according to Appendix E.3). SCION and UNCONSTRAINED SCION use the (Sign $\to$ Spectral $\to$ Sign) configuration with scaling factors in accordance with Tables 3 and 4. We train for 5100 iterations with a batch-size of 512 on the FineWeb dataset (see Table 7 regarding hyperparameters). In comparison with Adam, both Muon and (UNCONSTRAINED) SCION do not require learning rate warmup. We sweep over stepsizes and model width in Figure 1.

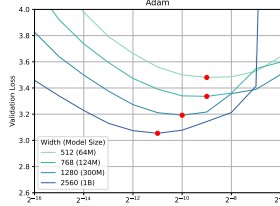 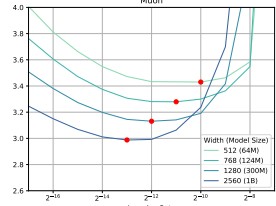 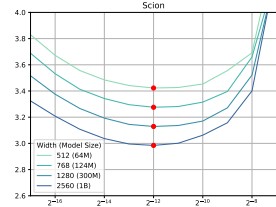 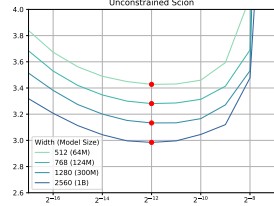

*Figure 1.* Performance on NanoGPT with between 64M and 1B parameters. The optimal learning rate of SCION is invariant to width.

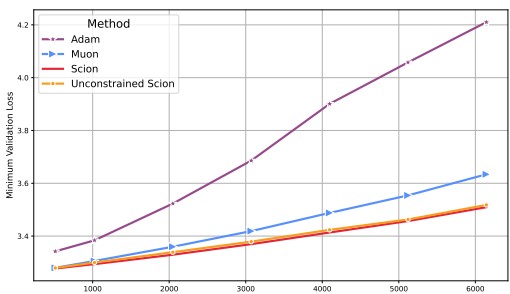

*Figure 2.* Batch size sensitivity on NanoGPT (124M). The generalization of SCION is less sensitive to larger batches.

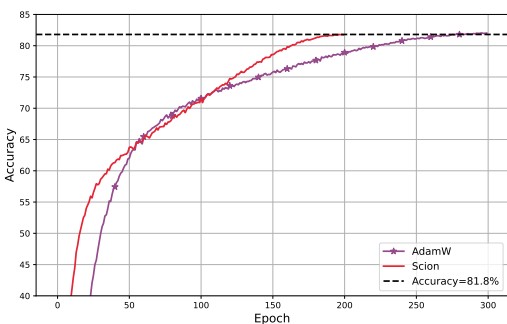

*Figure 3.* SCION leads to 30% fewer epochs for ViT on ImageNet and >40% wallclock speedup due to a larger critical batch size.

From Figure 1, we observe that the optimal stepsize of SCION and UNCONSTRAINED SCION transfer across model width as oppose to Adam and Muon. The 124M model size configuration in Figure 1 corresponds to one of the official speedrun entries of Jordan et al. (2024a), for which SCION has slightly lower validation loss (across 3 runs) than Muon. Even when Muon is tuned on the largest model size it achieves a validation loss of 2.988 in comparison with 2.984 of UNCONSTRAINED SCION.

Our methods completely remove the need for using Adam otherwise present in the Muon implementation, which permits an implementation that only requires storing one set of weights and one set of gradient (stored in half-precision) across all layers (see Appendix E.2). The experiments additionally demonstrates that our method works for weight sharing.

**3B model** Using the optimal configuration of the 124M parameter proxy model, we perform a large model experiment on a 3B parameter model, which also increases the depth. Specifically, we take the embedding dimension to be 2560 and the depth to be 36. We observe in Table 5 that UNCONSTRAINED SCION outperforms all other methods. The loss curve is provided in Figure 8 of Appendix E.

*Table 5.* Validation loss on a 3B parameter GPT model.

| Adam | Muon | UNCONSTRAINED SCION | SCION |
|------|------|---------------------|-------|
| 3.024 | 2.909 | **2.882** | 2.890 |

**Large batches** To test the effect of large batches we fix the total number of tokens for the 124M parameter model and sweep over the batch sizes while rescaling the total number of steps accordingly. The stepsize $\gamma$ is optimized over $\{2^{-17}, 2^{-16}, ..., 2^{-5}\}$ for each combination of batch size and optimizer. We observe that (UNCONSTRAINED) SCION is better at maintaining a low validation loss with increasing batch size than the baselines (*cf.*, Figure 2).

For large batches, SCION achieves a significantly better validation loss than Muon. To assess the implication for training time, we perform an additional experiment for the large batch size of 6144, where we reduce the number of iterations until SCION matches the larger validation loss of Muon. We find that SCION can achieve the same validation loss as Muon with a 25% smaller wallclock time.

We provide two possible explanations for why SCION favors large batches: The treatment of the noise is tied to the Euclidean geometry, in order to exploit the unbiasedness of the stochastic oracle, as apparent from the analysis. Additionally, in the extreme case of a single sample, the gradients of the linear layers are rank-1 and all Schatten norms become equivalent (e.g., Frobenius and Spectral norm).

**Image classification** We additionally test on vision transformers (ViT) on ImageNet and convolutional neural networks (CNN) on the CIFAR10 dataset using the configuration (Spectral → Spectral → Sign). The explicit control on the norm provided by SCION circumvents the need for the Frobenius norm normalization of the weights present in the CIFAR10 implementation of Muon (Jordan, 2024). The results regarding ImageNet are shown in Figure 3 (*cf.* Appendix E.4 for details and experiments on CIFAR10).

## Impact Statement

This paper presents work whose goal is to advance the field of Machine Learning. There are many potential societal consequences of our work, none which we feel must be specifically highlighted here.

## Acknowledgement

We thank Leena Chennuru Vankadara and Jeremy Bernstein for helpful discussion. This work was supported as part of the Swiss AI Initiative by a grant from the Swiss National Supercomputing Centre (CSCS) under project ID a06 on Alps. This work was supported by the Swiss National Science Foundation (SNSF) under grant number 200021_205011. This work was supported by Hasler Foundation Program: Hasler Responsible AI (project number 21043). Research was sponsored by the Army Research Office and was accomplished under Grant Number W911NF-24-1-0048. ASF was supported by a public grant from the Fondation Mathématique Jacques Hadamard.

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

# Appendix

## Table of Contents

# A. Preliminaries

## A.1. Relationship between steepest descent and uSCG

There are two prominent families of norm-based non-Euclidean method, namely the ones based on the lmo and the ones based on the sharp operator, both of which can be expressed in the terms of the Fenchel conjugate. The Fenchel conjugate of a proper, convex, and lower semicontinuous function $h : \mathcal{X} \to \mathbb{R} \cup \{\infty\}$ is defined as:

$$h^*(s) = \sup_{x \in \mathcal{X}} \{\langle s, x \rangle - h(x)\},$$

where $s \in \mathcal{X}$. The subdifferential $\partial h^*$ is equivalent to the argmax of the conjugate operation, i.e.,

$$\partial h^*(s) = \underset{x \in \mathcal{X}}{\operatorname{argmax}} \{\langle s, x \rangle - h(x)\}.$$

This follows from the Fenchel-Young inequality (see e.g. Bauschke & Lucet (2012)).

**LMO**   The lmo is a special case when $h$ is an indicator function of a convex set, i.e.,

$$\operatorname{lmo}(s) = \partial h^*(-s)$$

$$\text{with} \quad h(x) = \iota_{\mathcal{D}}(x) := \begin{cases} 0 & x \in \mathcal{D} \\ +\infty & \text{otherwise} \end{cases}$$

The lmo is commonly used for constrained minimization in e.g., CG since the operator ensure feasibility on the constrained set $\mathcal{D}$. When $\mathcal{D} := \{x \mid \|x\| \le \rho\}$, the lmo satisfies $\langle s, \operatorname{lmo}(s)\rangle = -\rho\|s\|_*$, which is central to the convergence proof of uSCG (see Lemma D.1).

**Sharp operator**   Another important example is the sharp operator (Nesterov, 2012; Kelner et al., 2014) defined as

$$s^\sharp \in \underset{x \in \mathcal{X}}{\arg\max}\{\langle s, x \rangle - \tfrac{1}{2}\|x\|^2\}$$

for some norm $\|\cdot\|$, which can equivalently be written as

$$s^\sharp \in \partial h^*(s) \quad \text{with} \quad h(x) = \tfrac{1}{2}\|x\|^2.$$

The sharp operator satisfies $\langle s, s^\sharp\rangle = \|s^\sharp\|^2 = \|s\|_*^2$ (Kelner et al., 2014, App. A.1).

The sharp operator and lmo can be defined in terms of each other when $\mathcal{D} := \{x \mid \|x\| \le \rho\}$, specifically

$$s^\sharp = -\tfrac{1}{\rho}\|s\|_* \operatorname{lmo}(s) \tag{9}$$

From (9) we see a clear distinction between the lmo and the sharp operator, namely that, while the lmo is scale invariant (i.e. $\operatorname{lmo}(a \cdot s) = \operatorname{lmo}(s)$ for $a > 0$) the sharp operator is not (since $[a \cdot s]^\sharp = a[s]^\sharp$ for $a \in \mathbb{R}$).

**Steepest descent**   Steepest descent in a normed space can be written in terms of the sharp operator as follows

$$x^{k+1} = x^k - \gamma[\nabla f(x^k)]^\sharp$$

with a stepsize $\gamma > 0$. From (9) it becomes apparent that uSCG can be seen as a normalized variant of steepest descent with momentum.

# B. Method

## B.1. Input radius scaling

Based on the spectral norm perspective (Yang et al., 2023), which requires that $\|W_\ell\|_{\mathcal{S}_\infty} = \Theta(\sqrt{d_{\text{out}}/d_{\text{in}}})$, one might be inclined to pick the initialization such that $\|W_\ell\|_{\mathcal{S}_\infty} = \sqrt{d_{\text{out}}/d_{\text{in}}}$ is ensured exactly. This argument is indeed valid

---

**Algorithm 3** (Unconstrained) Scion

---

**Input:** Horizon $n$, init. $x^1 = (W_1^1, ..., W_L^1)$, $d^0 = 0$, momentum $\alpha_k \in (0, 1]$, stepsize $\gamma \in (0, 1)$, radii $\rho_i \in \mathbb{R}_+$.

1: **for** $k = 1, \ldots, n - 1$ **do**
2:      Sample $\xi_k \sim \mathcal{P}$
3:      $d^k \leftarrow \alpha_k \nabla f(x^k, \xi_k) + (1 - \alpha_k) d^{k-1}$
4:      $x_\ell^{k+1} \leftarrow \begin{cases} x_\ell^k + \gamma \rho_\ell \, \mathrm{lmo}_{\|\cdot\|_{\alpha_\ell \to \beta_\ell}}(d_\ell^k) & \textbf{if} \text{ unconstrained} \\ (1 - \gamma) x_\ell^k + \gamma \rho_\ell \, \mathrm{lmo}_{\|\cdot\|_{\alpha_\ell \to \beta_\ell}}(d_\ell^k) & \textbf{else} \end{cases} \qquad \forall \ell \in [L]$

**Return** $x^n$

---

SCG and uSCG with the layerwise norm choice from (6). For simplicity we ignore biases.

---

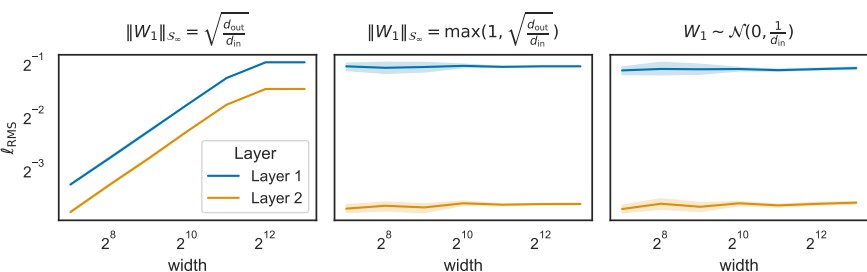

*Figure 4.* Coordinate check at initialization. Preactivations are not constant with the spectral scaling $\sqrt{\frac{d_{\mathrm{out}}}{d_{\mathrm{in}}}}$, when $d_{\mathrm{in}} > d_{\mathrm{out}}$.

asymptotically, since input dimension is kept fixed. However, when the input dimension is larger than the output dimension, this does not lead to constant preactivations as demonstrated through a coordinate check (Yang & Hu, 2021) carried out in Figure 4.

Kaiming initialization (He et al., 2015) fortunately circumvents this problem. From random matrix theory we have that $\|A\|_{\mathcal{S}_\infty} \approx \sigma(\sqrt{d_{\mathrm{in}}} + \sqrt{d_{\mathrm{out}}})$ for $A_{ij} \sim \mathcal{N}(0, \sigma^2)$ (Vershynin, 2018). So the Kaiming initialization, $[W_\ell]_{ij} \sim N(0, 1/d_{\mathrm{in}})$, leads to $\|W_\ell\|_{\mathcal{S}_\infty} \approx 1 + \sqrt{d_{\mathrm{out}}/d_{\mathrm{in}}}$, which prevents the preactivation from going to zero as $d_{\mathrm{out}} \to 0$. Alternatively, one can simply choose $\|W_\ell\|_{\mathcal{S}_\infty} = \max(1, \sqrt{d_{\mathrm{out}}/d_{\mathrm{in}}})$.

Ensuring a correct norm scaling is particularly important for SCG and uSCG, since the scaling not only affects initialization but also the update rule itself. Specifically, if the methods were run with the norm bound choice $\|W_\ell\|_{\mathcal{S}_\infty} \leq \sqrt{d_{\mathrm{out}}/d_{\mathrm{in}}}$ for the input layer, then the issue in Figure 4 persists, due the lmo always lying on the boundary of the norm ball. The choice $\|W_\ell\|_{\mathcal{S}_\infty} = \max(1, \sqrt{d_{\mathrm{out}}/d_{\mathrm{in}}})$ resolves this issue.

## B.2. Alternative norm choices

The primary argument in Section 3.1 for the norm choice is based on the invariance provided by the $\mathrm{RMS} \to \mathrm{RMS}$ operator norm: i.e., the RMS norm of the output of layer $\ell$ is bounded, so the input of next layer $\ell + 1$ is also bounded in the RMS norm. Since the lmo of $\|\cdot\|_{\mathrm{RMS}\to\mathrm{RMS}}$ can be computed efficiently, we can directly use this norm choice for our update rule.

However, it is possible to choose another norm such as $\|\cdot\|_{1\to\mathrm{RMS}}$, as long as the RMS norm guarantee on the output of $W_\ell$ is converted into a guarantee on the $\ell_1$-norm of the input of layer $\ell + 1$. Specifically, we have that $\|\cdot\|_{\mathrm{RMS}\to\mathrm{RMS}} \leq d_{\mathrm{in}}\|\cdot\|_{1\to\mathrm{RMS}}$ through Fact 3.1. Alternatively, we can rely on the invariance provided by $\|\cdot\|_{\infty\to\infty}$, for which Fact 3.1 tells us that $\|\cdot\|_{\infty\to\infty} \leq \sqrt{d_{\mathrm{in}}}\|\cdot\|_{\mathrm{RMS}\to\infty}$ and $\|\cdot\|_{\infty\to\infty} \leq d_{\mathrm{in}}\|\cdot\|_{1\to\infty}$. The resulting lmo choices for the three norm choices across all layers are summarized in Table 6.

*Table 6.* It is possible to use the same norm throughout the network if scaled appropriately. By not treating the network as a flattened vector, hyperparameter can transfer across model sizes (cf. Figure 6). We have made use of Fact 3.1 and (5) to derive the correct layerwise scaling. We assume that the image input dimension is smaller than the $d_{\text{out}}$ of the first layer (otherwise see Appendix B.1).

| Weight norm (bias norm) | | | $W_1$ (1-hot encoded) | $W_1$ (image domain) | $(W_\ell)_{\ell \in [2,\ldots,L-1]}$ | $W_L$ | $b_\ell$ |
|---|---|---|---|---|---|---|---|
| Spectral | RMS $\to$ RMS (RMS) | lmo | $-\sqrt{d_{\text{out}}}UV^\top$ | $-\sqrt{d_{\text{out}}/d_{\text{in}}}UV^\top$ | | | $-\frac{b_\ell}{\|b_\ell\|_{\text{RMS}}}$ |
| ColNorm | $1 \to$ RMS (RMS) | lmo | $\text{col}_j(W_1) \mapsto -\sqrt{d_{\text{out}}}\frac{\text{col}_j(W_1)}{\|\text{col}_j(W_1)\|_2}$ | $\text{col}_j(W_\ell) \mapsto -\frac{\sqrt{d_{\text{out}}}}{d_{\text{in}}}\frac{\text{col}_j(W_\ell)}{\|\text{col}_j(W_\ell)\|_2}$ | | | $-\frac{b_\ell}{\|b_\ell\|_{\text{RMS}}}$ |
| RowNorm | RMS $\to \infty$ (RMS) | lmo | $\text{row}_i(W_1) \mapsto -\frac{\text{row}_i(W_1)}{\|\text{row}_i(W_1)\|_2}$ | $\text{row}_i(W_\ell) \mapsto -\frac{\text{row}_i(W_\ell)}{\sqrt{d_{\text{in}}}\|\text{row}_i(W_\ell)\|_2}$ | | | $-\frac{b_\ell}{\|b_\ell\|_{\text{RMS}}}$ |
| Sign | $1 \to \infty$ ($\infty$) | lmo | $-\text{sign}(W_1)$ | $-\frac{1}{d_{\text{in}}}\text{sign}(W_\ell)$ | | | $-\text{sign}(b_\ell)$ |

## B.3. Boundary initialization

**Semi-orthogonal**   Following Saxe et al. (2013), perform QR decomposition of a random matrix

$$G_{ij} \sim \mathcal{N}(0,1), \quad \forall i,j$$
$$G = QR$$

Use $Q' = Q\,\text{sign}(\text{diag}(R))$ as the semi-orthogonal matrix as the initialization.

**Column-wise normalized Gaussian**   As proposed in Large et al. (2024), initialize each column as follows

$$W_{ij} \sim \mathcal{N}(0,1), \quad \forall i,j$$
$$\text{col}_j(W) = \frac{\text{col}_j(W)}{\|\text{col}_j(W)\|_2}, \quad \forall i$$

**Row-wise normalized Gaussian**   Initialize each row as follows

$$W_{ij} \sim \mathcal{N}(0,1), \quad \forall i,j$$
$$\text{row}_i(W) = \frac{\text{row}_i(W)}{\|\text{row}_i(W)\|_2}, \quad \forall j$$

**Random sign**

$$W_{ij} = \begin{cases} +1 & \text{with probability } 0.5 \\ -1 & \text{with probability } 0.5 \end{cases} \quad \forall i,j$$

Each initialization should be scaled by the corresponding scaling of the lmo elementwise.

## C. Proofs for Section 3 (Our Methods)

### C.1. Notation and detailed problem setting

First, we introduce some notation and detailed problem settings. Consider an $L$-layer neural network with input dimension $d_0$, hidden layer widths $\{d_1, d_2, \ldots, d_{L-1}\}$, and output dimension $d_L$. For any input data $\mathbf{z} \in \mathbb{R}^{d_0}$, the forward pass of the network is defined as

$$\begin{aligned} \mathbf{f}^{(1)}(\mathbf{z}) &= \mathbf{W}^{(1)}\mathbf{z}, & \mathbf{h}^{(1)}(\mathbf{z}) &= \sigma\big(\mathbf{f}^{(1)}(\mathbf{z})\big), \\ \mathbf{f}^{(\ell)}(\mathbf{z}) &= \mathbf{W}^{(\ell)}\mathbf{h}^{(\ell-1)}(\mathbf{z}), & \mathbf{h}^{(\ell)}(\mathbf{z}) &= \sigma\big(\mathbf{f}^{(\ell)}(\mathbf{z})\big), & \forall \ell = 2, \ldots, L-1, \\ \mathbf{f}^{(L)}(\mathbf{z}) &= \mathbf{W}^{(L)}\mathbf{h}^{(\ell-1)}(\mathbf{z}). \end{aligned} \tag{10}$$

Here, $\mathbf{W}^{(\ell)} \in \mathbb{R}^{d_\ell \times d_{\ell-1}}$ are the weight matrices of the network, and $\sigma(\cdot)$ is an element-wise activation function. We denote $\mathbf{f}^{(\ell)}(\mathbf{z}) \in \mathbb{R}^{d_\ell}$ as the preactivation of the $l$-th layer and $\mathbf{h}^{(\ell)}(\mathbf{z}) \in \mathbb{R}^{d_\ell}$ as the corresponding postactivation. The network output $\mathbf{f}^{(L)}(\mathbf{z})$ is calculated as a linear transformation of the final hidden layer representation.

Below, we provide a brief overview of the specific assumptions for the input data and initialization. Then we explicitly define the spectral norm choice used in our analysis:

**Assumption C.1** (Input data). *Training samples* $(\mathbf{z}, \mathbf{y})$ *are drawn from a distribution* $\mathcal{P}$, *where* $\mathbf{z} \in \mathbb{R}^{d_0}$ *and* $\mathbf{y} \in \mathbb{R}^{d_L}$. *We assume* $\mathbf{z}$ *has bounded second moments, i.e.,* $\|\mathbf{z}\|_2^2 < \infty$.

**Assumption C.2** (Initialization schemes). *The initialization satisfy the following condition for the stability of the maximal update learning rate ([Yang et al., 2023](#)):*

$$\|\mathbf{W}^{(\ell)}\|_{\mathcal{S}_\infty} = \Theta\left(\sqrt{\tfrac{d_\ell}{d_{\ell-1}}}\right).$$

*Both semi-orthogonal initialization and (with high probability) Gaussian initialization satisfy this condition.*

**Assumption C.3** (Spectral norm choice for lmo). *We adopt the layer-wise linear maximization oracles (lmos) from Tables [3](#) and [4](#). Specifically, for the first layer* $\mathbf{W}^{(1)}$, *we use*

$$\mathrm{lmo}\left(\nabla_{\mathbf{W}^{(1)}} \mathcal{L}\right) \;=\; \max\left(1, \sqrt{\tfrac{d_{\mathrm{out}}}{d_{\mathrm{in}}}}\right) \mathbf{U}^{(1)} \mathbf{V}^{(1)\top},$$

*where* $\mathbf{W}^{(1)} = \mathbf{U}^{(1)} \mathbf{\Lambda}^{(1)} \mathbf{V}^{(1)\top}$ *is the reduced SVD of* $\mathbf{W}^{(1)}$. *For each intermediate layer* $\mathbf{W}^{(\ell)}$, $\ell \in \{2, \dots, L\}$, *we similarly set*

$$\mathrm{lmo}\left(\nabla_{\mathbf{W}^{(\ell)}} \mathcal{L}\right) \;=\; \sqrt{\tfrac{d_{\mathrm{out}}}{d_{\mathrm{in}}}} \mathbf{U}^{(\ell)} \mathbf{V}^{(\ell)\top},$$

*Remark* C.4. In all cases, these choices are consistent with the (Spectral → Spectral → Spectral) configuration from Tables [3](#) and [4](#). Our results will simultaneously hold for the (ColNorm → Spectral → Spectral) configuration, due to the equivalence under 1-hot encoded (*cf.* Insight [3.2](#)).

To investigate how these neuron preactivations change after one step of Algorithm [1](#), we consider a batch size $= 1$ setting. We analyze the update dynamics of the network parameters under general loss functions $\mathcal{L}$, including but not limited to mean squared error (MSE) and logistic loss.

We follow [Yang & Hu (2021)](#) which states that a *good* learning rate enables hidden layer preactivations to undergo the largest possible change in a single update step, while still avoiding divergence when the network width is large.

Let $\Delta f_i^{(\ell)}(\mathbf{z})$ be the change in the preactivation of the $i$-th neuron in the $l$-th hidden layer after one step of Algorithm [1](#) on $(\mathbf{z}, \mathbf{y})$. The so-called "maximal update" heuristic requires that:

$$\text{The maximal update learning rate } \gamma^* \;:=\; \text{the learning rate for which } \mathbb{E}\left[\left(\Delta f_i^{(\ell)}(\mathbf{z})\right)^2\right] \;\simeq\; 1,$$

with the expectation again taken over the initialization distribution.

## C.2. Hyperparameter transfer

To begin with, we prove the following lemma that derives the lmo of the gradient in an $L$-layer neural network using the spectral norm choice from Table [3](#).

**Lemma C.5** (Spectral lmo for the gradient with respect to $\mathbf{W}^{(\ell)}$). *Consider an $L$-layer neural network with input dimension $d_0$, hidden layer widths $\{d_1, d_2, \dots, d_{L-1}\}$, and output dimension $d_L$. Training samples $(\mathbf{z}, \mathbf{y})$ are drawn from some distribution $\mathcal{P}$, where $\mathbf{z} \in \mathbb{R}^{d_0}$ and $\mathbf{y} \in \mathbb{R}^{d_L}$. The network follows the forward pass ([10](#)). For convenience, we set $\mathbf{f}^{(0)} = \mathbf{h}^{(0)} = \mathbf{z}$, making the notation consistent for all layers. The linear maximization oracle (lmo) over the scaled spectral norm ball, $\mathcal{D} = \left\{ \mathbf{W} \mid \|\mathbf{W}\|_{\mathcal{S}_\infty} \leq \sqrt{\tfrac{d_\ell}{d_{\ell-1}}} \right\}$, for $\nabla_{\mathbf{W}^{(\ell)}} \mathcal{L}(\mathbf{z}, \mathbf{y})$, denoted as $\mathrm{lmo}(\nabla_{\mathbf{W}^{(\ell)}} \mathcal{L}(\mathbf{z}, \mathbf{y}))$, is given by*

$$\mathrm{lmo}(\nabla_{\mathbf{W}^{(\ell)}} \mathcal{L}(\mathbf{z}, \mathbf{y})) = \sqrt{\frac{d_\ell}{d_{\ell-1}}} \frac{\left(\frac{d\mathcal{L}(\mathbf{z},\mathbf{y})}{d\mathbf{f}^{(\ell)}}\right) \mathbf{h}^{(\ell-1)\top}}{\left\|\frac{d\mathcal{L}(\mathbf{z},\mathbf{y})}{d\mathbf{f}^{(\ell)}}\right\|_2 \|\mathbf{h}^{(\ell-1)}\|_2}.$$

*Proof.* First, we express the gradient with respect to $\mathbf{W}^{(\ell)}$. By applying the chain rule we have

$$\nabla_{\mathbf{W}^{(\ell)}} \mathcal{L}(\mathbf{z}, \mathbf{y}) = \left(\frac{d\mathcal{L}(\mathbf{z}, \mathbf{y})}{d\mathbf{f}^{(\ell)}}\right) \mathbf{h}^{(\ell-1)\top}.$$

We immediately have that the lmo is given as

$$\text{lmo}(\nabla_{\mathbf{W}^{(\ell)}}\mathcal{L}(\mathbf{z},\mathbf{y})) = \sqrt{\frac{d_\ell}{d_{\ell-1}}} \frac{\left(\frac{d\mathcal{L}(\mathbf{z},\mathbf{y})}{d\mathbf{f}^{(\ell)}}\right)\mathbf{h}^{(\ell-1)\top}}{\left\|\frac{d\mathcal{L}(\mathbf{z},\mathbf{y})}{d\mathbf{f}^{(\ell)}}\right\|_2 \|\mathbf{h}^{(\ell-1)}\|_2}.$$

$\square$

Now, we are equipped to state and prove Lemma C.6. We follow the proof technique of Yang et al. (2023) in our derivations.

**Lemma C.6** (Width-invariance of the maximal update learning rate). *Consider an L-layer MLP with widths $d_0, d_1, \ldots, d_L$ (where $d_1 \geq d_0$), and assume $d_0$ is a fixed constant. Let its activation function $\sigma$ have Lipschitz constant $L_\sigma$ and satisfy $\sigma(0) = 0$ (e.g., ReLU, Tanh, or GELU). Suppose:*

 *(i) The input data $(\mathbf{z}, \mathbf{y})$ meets the requirements in Assumption C.1,*

 *(ii) The network is initialized according to Assumption C.2, and*

*(iii) Parameter updates use Algorithm 1 with the spectral-norm-based lmo described in Assumption C.3.*

*For various loss functions $\mathcal{L}$ (e.g., MSE, logistic), define the* maximal update condition *by*

$$\mathbb{E}\left[\left(\Delta f_i^{(\ell)}(\mathbf{z})\right)^2\right] \simeq 1 \quad \text{for all } \ell \leq L,$$

*where $\Delta f_i^{(\ell)}(\mathbf{z})$ is the change in the preactivation of the $i$-th neuron in the $\ell$-th hidden layer after one update step. Unless all activations are simultaneously zero during training (which is highly unlikely in practice), the optimal learning rate $\gamma^*$ (under the setting $\alpha_k = 1$ in Algorithm 1) satisfying this condition is* independent of the hidden-layer widths.

*Proof.* For Algorithm 1 with the setting $\alpha_k = 1$, we have through the spectral norm choice Assumption C.3 and Lemma C.5 that

$$\begin{aligned}
\Delta f_i^{(\ell)}(\mathbf{z}) &= \sum_{j=1}^{d_{\ell-1}} \Delta W_{i,j}^{(\ell)} h_j^{(\ell-1)}(\mathbf{z}) + \sum_{j=1}^{d_{\ell-1}} W_{i,j}^{(\ell)} \Delta h_j^{(\ell-1)}(\mathbf{z}) \\
&= \gamma \sum_{j=1}^{d_{\ell-1}} \left[\text{lmo}(\nabla_{\mathbf{W}^{(\ell)}}\mathcal{L}(\mathbf{z},\mathbf{y}))\right]_{i,j} h_j^{(\ell-1)}(\mathbf{z}) + (\mathbf{W}^{(\ell)}\Delta\mathbf{h}^{(\ell-1)})_i \\
&= \gamma \sum_{j=1}^{d_{\ell-1}} \sqrt{\frac{d_\ell}{d_{\ell-1}}} \frac{\left(\frac{d\mathcal{L}(\mathbf{z},\mathbf{y})}{d\mathbf{f}^{(\ell)}}\right)_i}{\left\|\frac{d\mathcal{L}(\mathbf{z},\mathbf{y})}{d\mathbf{f}^{(\ell)}}\right\|_2} \frac{h_j^{(\ell-1)}(\mathbf{z})}{\|\mathbf{h}^{(\ell-1)}(\mathbf{z})\|_2} h_j^{(\ell-1)}(\mathbf{z}) + (\mathbf{W}^{(\ell)}\Delta\mathbf{h}^{(\ell-1)})_i \\
&= \gamma \sqrt{\frac{d_\ell}{d_{\ell-1}}} \frac{\left(\frac{d\mathcal{L}(\mathbf{z},\mathbf{y})}{d\mathbf{f}^{(\ell)}}\right)_i}{\left\|\frac{d\mathcal{L}(\mathbf{z},\mathbf{y})}{d\mathbf{f}^{(\ell)}}\right\|_2} \|\mathbf{h}^{(\ell-1)}(\mathbf{z})\|_2 + (\mathbf{W}^{(\ell)}\Delta\mathbf{h}^{(\ell-1)})_i.
\end{aligned}$$

Unless these terms perfect cancel each other out, we have

$$\begin{aligned}
\mathbb{E}\left[\|\Delta\mathbf{f}^{(\ell)}(\mathbf{z})\|_2^2\right] &= \mathbb{E}\sum_{i=1}^{d_\ell}\left[\left(\Delta f_i^{(\ell)}(\mathbf{z})\right)^2\right] \\
&\simeq \mathbb{E}\sum_{i=1}^{d_\ell}\left[\gamma^2 \frac{d_\ell}{d_{\ell-1}} \frac{\left(\frac{d\mathcal{L}(\mathbf{z},\mathbf{y})}{d\mathbf{f}^{(\ell)}}\right)_i^2}{\left\|\frac{d\mathcal{L}(\mathbf{z},\mathbf{y})}{d\mathbf{f}^{(\ell)}}\right\|_2^2} \|\mathbf{h}^{(\ell-1)}(\mathbf{z})\|_2^2\right] + \mathbb{E}\sum_{i=1}^{d_\ell}\left[(\mathbf{W}^{(\ell)}\Delta\mathbf{h}^{(\ell-1)})_i^2\right] \\
&= \gamma^2 \frac{d_\ell}{d_{\ell-1}} \mathbb{E}\|\mathbf{h}^{(\ell-1)}(\mathbf{z})\|_2^2 + \mathbb{E}\left\|\mathbf{W}^{(\ell)}\Delta\mathbf{h}^{(\ell-1)}\right\|_2^2.
\end{aligned}$$

For the second term, we have

$$\mathbb{E}\|\Delta \mathbf{h}^{(\ell-1)}(\mathbf{z})\|_2^2 \le L_\sigma^2 \mathbb{E}\left\|\Delta \mathbf{f}^{(\ell-1)}(\mathbf{z})\right\|_2^2$$

So

$$\mathbb{E}\left\|\mathbf{W}^{(\ell)}\Delta \mathbf{h}^{(\ell-1)}\right\|_2^2 \le \frac{d_\ell}{d_{\ell-1}}L_\sigma^2 \mathbb{E}\left\|\Delta \mathbf{f}^{(\ell-1)}(\mathbf{z})\right\|_2^2$$

Recall that, under Assumption C.1, the input $\mathbf{z}$ has bounded second moments, i.e., $\|\mathbf{z}\|_2^2 < \infty$. Consequently, we can treat $\|\mathbf{z}\|_2^2$, $d_0$, and $L_\sigma$ as constants. Under these conditions, we have:

$$\mathbb{E}\big[\|\Delta \mathbf{f}^{(\ell)}(\mathbf{z})\|_2^2\big] \simeq \gamma^2 \frac{d_\ell}{d_{\ell-1}}\mathbb{E}\|\mathbf{h}^{(\ell-1)}(\mathbf{z})\|_2^2 + \mathbb{E}\left\|\mathbf{W}^{(\ell)}\Delta \mathbf{h}^{(\ell-1)}\right\|_2^2,$$

$$\mathbb{E}\left\|\mathbf{W}^{(\ell)}\Delta \mathbf{h}^{(\ell-1)}\right\|_2^2 = \frac{d_\ell}{d_{\ell-1}}\mathcal{O}\left(\mathbb{E}\left\|\Delta \mathbf{f}^{(\ell-1)}(\mathbf{z})\right\|_2^2\right),$$

$$\mathbb{E}\|\Delta \mathbf{h}^{(\ell-1)}(\mathbf{z})\|_2^2 = \mathcal{O}\left(\mathbb{E}\left\|\Delta \mathbf{f}^{(\ell-1)}(\mathbf{z})\right\|_2^2\right),$$

Recall that, by Assumption C.1, the input $\mathbf{z}$ has bounded second moments, i.e., $\mathbb{E}[\|\mathbf{z}\|_2^2] < \infty$. Focusing on the case $\ell = 1$, we obtain

$$\mathbb{E}\big[\|\Delta \mathbf{f}^{(1)}(\mathbf{z})\|_2^2\big] = \frac{\gamma^2 d_1 \|\mathbf{z}\|_2^2}{d_0} \simeq \gamma^2 d_1 .$$

Moreover, by the Assumption C.2 and leveraging the proof from Yang et al. (2023) (specifically, by Eq. (8) at initialization), we have:

$$\mathbb{E}\|\mathbf{h}^{(\ell)}(\mathbf{z})\|_2^2 = \Theta(d_\ell) \quad \forall \ell \in [L-1] .$$

By induction, unless all activations are simultaneously zero during training (which is unlikely in practice), we have:

$$\mathbb{E}\big[\|\Delta \mathbf{f}^{(\ell)}(\mathbf{z})\|_2^2\big] \simeq \gamma^2 d_\ell .$$

By symmetry, we obtain:

$$\mathbb{E}\big[\|\Delta f_i^{(\ell)}(\mathbf{z})\|_2^2\big] = \frac{1}{d_\ell}\mathbb{E}\big[\|\Delta \mathbf{f}^{(\ell)}(\mathbf{z})\|_2^2\big] \simeq \gamma^2 \quad \forall i \in [d_\ell],$$

where independent with the width for every hidden layer. $\square$

*Remark* C.7. The maximal update condition ensures that the network operates in a *stable but maximally adaptive regime*, balancing *efficient learning and numerical stability*.

*Remark* C.8. As the hidden layer widths $d_1, \ldots, d_{L-1}$ increase, the learning rate required to maintain the maximal update property remains unchanged, demonstrating width invariance in deep networks. Consequently, a learning rate tuned on a smaller model can be directly applied to a wider model without sacrificing training dynamics.

# D. Proofs for Section 5 (Analysis)

In this section we present the proofs of the main convergence results of the paper as well as some intermediary lemmas that we will make use of along the way. Throughout this section, we adopt the notation:

$$
\begin{aligned}
\text{(stochastic gradient estimator error)} \quad & \lambda^k := d^k - \nabla f(x^k) \\
\text{(diameter of } \mathcal{D} \text{ in } \ell_2 \text{ norm)} \quad & D_2 := \max_{x,y \in \mathcal{D}} \|x - y\|_2 \\
\text{(radius of } \mathcal{D} \text{ in } \ell_2 \text{ norm)} \quad & \rho_2 := \max_{x \in \mathcal{D}} \|x\|_2 \\
\text{(norm equivalence constant)} \quad & \zeta := \max_{x \in \mathcal{X}} \frac{\|x\|_*}{\|x\|_2} \\
\text{(Lipschitz constant of } \nabla f \text{ with respect to } \|\cdot\|_2) \quad & L_2 := \inf\{M > 0 \colon \forall x, y \in \mathcal{X}, \|\nabla f(x) - \nabla f(y)\|_2 \le M \|x - y\|_2\}
\end{aligned}
$$

We analyze each algorithm separately, although the analysis is effectively unified between the two, modulo constants. This is done in Appendices D.1 and D.2, respectively. Our convergence analysis proceeds in three steps: we begin by establishing a template descent inequality for each algorithm via the descent lemma. Next, we analyze the behavior of the second moment of the error $\mathbb{E}[\|\lambda^k\|_2^2]$ under different choices for $\alpha$. Then, we combine these results to derive a convergence rate. Finally, we note that when analyzing algorithms with constant momentum, we will still always take $\alpha = 1$ on the first iteration $k = 1$.

## D.1. Convergence analysis of uSCG

We begin with the analysis of Algorithm 1 by establishing a generic template inequality for the dual norm of the gradient at iteration $k$. This inequality holds regardless of whether the momentum $\alpha_k$ is constant or vanishing, as long as it remains in $(0, 1]$.

**Lemma D.1** (uSCG template inequality). *Suppose Assumption 5.1 holds. Let $n \in \mathbb{N}^*$ and consider the iterates $\{x^k\}_{k=1}^n$ generated by Algorithm 1 with a constant stepsize $\gamma > 0$. Then we have*

$$
\mathbb{E}[\|\nabla f(\bar{x}^n)\|_*] \le \frac{\mathbb{E}[f(x^1) - f^\star]}{\rho \gamma n} + \frac{L \rho \gamma}{2} + \frac{1}{n}\left(\frac{\rho_2}{\rho} + \zeta\right) \sum_{k=1}^n \sqrt{\mathbb{E}[\|\lambda^k\|_2^2]}. \tag{11}
$$

*Proof.* Under Assumption 5.1, we can use the descent lemma for the function $f$ at the points $x^k$ and $x^{k+1}$ to get, for all $k \in \{1, \dots, n\}$,

$$
\begin{aligned}
f(x^{k+1}) &\le f(x^k) + \langle \nabla f(x^k), x^{k+1} - x^k \rangle + \tfrac{L}{2} \|x^{k+1} - x^k\|^2 \\
&= f(x^k) + \langle \nabla f(x^k) - d^k, x^{k+1} - x^k \rangle + \langle d^k, x^{k+1} - x^k \rangle + \tfrac{L}{2} \|x^{k+1} - x^k\|^2 \\
&= f(x^k) + \gamma \langle \nabla f(x^k) - d^k, \mathrm{lmo}(d^k) \rangle + \gamma \langle d^k, \mathrm{lmo}(d^k) \rangle + \tfrac{L\gamma^2}{2} \|\mathrm{lmo}(d^k)\|^2 \\
&\le f(x^k) + \gamma \rho_2 \|\lambda^k\|_2 + \gamma \langle d^k, \mathrm{lmo}(d^k) \rangle + \tfrac{L\gamma^2}{2} \rho^2,
\end{aligned} \tag{12}
$$

the final step employing Cauchy-Schwarz, the definition of $\lambda^k$, and the definition of $\rho_2$ as the radius of $\mathcal{D}$ in the $\|\cdot\|_2$ norm. By definition of the dual norm we have, for all $u \in \mathcal{X}$,

$$
\|u\|_* = \max_{v \colon \|v\| \le 1} \langle u, v \rangle = \max_{v \in \mathcal{D}} \langle u, \tfrac{1}{\rho} v \rangle = -\langle u, \tfrac{1}{\rho} \mathrm{lmo}(u) \rangle
$$

which means that, for all $k \in \{1, \dots, n\}$,

$$
\gamma \langle d^k, \mathrm{lmo}(d^k) \rangle = \gamma \rho \langle d^k, \tfrac{1}{\rho} \mathrm{lmo}(d^k) \rangle = -\gamma \rho \|d^k\|_*.
$$

Plugging this expression for $\gamma\langle d^k, \mathrm{lmo}(d^k)\rangle$ into (12) gives, for all $k \in \{1, \ldots, n\}$,

$$
\begin{aligned}
f(x^{k+1}) &\leq f(x^k) + \gamma\rho_2 \left\|\lambda^k\right\|_2 - \gamma\rho\|d^k\|_* + \tfrac{L\gamma^2}{2}\rho^2 \\
&= f(x^k) + \gamma\rho_2 \left\|\lambda^k\right\|_2 - \gamma\rho\|d^k - \nabla f(x^k) + \nabla f(x^k)\|_* + \tfrac{L\gamma^2}{2}\rho^2 \\
&\overset{(a)}{\leq} f(x^k) + \gamma\rho_2 \left\|\lambda^k\right\|_2 + \gamma\rho\|\lambda^k\|_* - \gamma\rho\|\nabla f(x^k)\|_* + \tfrac{L\gamma^2}{2}\rho^2 \\
&\overset{(b)}{\leq} f(x^k) + \gamma(\rho_2 + \zeta\rho) \left\|\lambda^k\right\|_2 - \gamma\rho\|\nabla f(x^k)\|_* + \tfrac{L\gamma^2}{2}\rho^2,
\end{aligned}
$$

applying the reverse triangle inequality in (a) while (b) stems from the definition of $\zeta$. By rearranging terms and taking expectations, we get

$$
\gamma\rho\mathbb{E}[\left\|\nabla f(x^k)\right\|_*] \leq \mathbb{E}[f(x^k) - f(x^{k+1})] + \gamma(\rho_2 + \zeta\rho)\mathbb{E}[\left\|\lambda^k\right\|_2] + \frac{L\rho^2\gamma^2}{2}.
$$

Summing this from $k = 1$ to $n$ and dividing by $\gamma\rho n$ we get

$$
\begin{aligned}
\mathbb{E}[\|\nabla f(\bar{x}^n)\|_*] &= \frac{1}{n}\sum_{k=1}^{n}\mathbb{E}[\left\|\nabla f(x^k)\right\|_*] \\
&\leq \frac{\mathbb{E}[f(x^1) - f(x^{n+1})]}{\rho\gamma n} + \frac{L\rho\gamma}{2} + \frac{1}{n}\left(\frac{\rho_2}{\rho} + \zeta\right)\sum_{k=1}^{n}\mathbb{E}[\left\|\lambda^k\right\|_2] \\
&\overset{(a)}{\leq} \frac{\mathbb{E}[f(x^1) - f^\star]}{\rho\gamma n} + \frac{L\rho\gamma}{2} + \frac{1}{n}\left(\frac{\rho_2}{\rho} + \zeta\right)\sum_{k=1}^{n}\mathbb{E}[\left\|\lambda^k\right\|_2] \\
&\overset{(b)}{\leq} \frac{\mathbb{E}[f(x^1) - f^\star]}{\rho\gamma n} + \frac{L\rho\gamma}{2} + \frac{1}{n}\left(\frac{\rho_2}{\rho} + \zeta\right)\sum_{k=1}^{n}\sqrt{\mathbb{E}[\|\lambda^k\|_2^2]},
\end{aligned}
$$

using the definition of $f^\star$ for (a) and Jensen's inequality for (b). $\qquad\square$

At this point, we need to determine the growth of the induced error captured by the quantity $\left\|\lambda^k\right\|_2^2$. To estimate this, we first use a recursion relating $\mathbb{E}[\left\|\lambda^k\right\|_2^2]$ and $\mathbb{E}[\left\|\lambda^{k-1}\right\|_2^2]$ adapted from the proof in Mokhtari et al. (2020, Lem. 6) and then we prove a bound on the decay of $\left\|\lambda^k\right\|_2^2$ for Algorithm 1.

**Lemma D.2** (Linear recursive inequality for $\mathbb{E}\left\|\lambda^k\right\|_2^2$). *Suppose Assumptions 5.1 and 5.3 hold. Let $n \in \mathbb{N}^*$ and consider the iterates $\{x_k\}_{k=1}^n$ generated by Algorithm 1 with a constant stepsize $\gamma > 0$. Then, for all $k \in \{1, \ldots, n\}$,*

$$
\mathbb{E}[\left\|\lambda^k\right\|_2^2] \leq \left(1 - \frac{\alpha_k}{2}\right)\mathbb{E}[\left\|\lambda^{k-1}\right\|_2^2] + \frac{2L_2^2\rho_2^2\gamma^2}{\alpha_k} + \alpha_k^2\sigma^2.
$$

*Proof.* The proof is a straightforward adaptation of the arguments laid out in Mokhtari et al. (2020, Lem. 6), which in fact do not depend on convexity nor on the choice of stepsize. Let $n \in \mathbb{N}^*$ and $k \in \{1, \ldots, n\}$, then

$$
\begin{aligned}
\left\|\lambda^k\right\|_2^2 &= \left\|\nabla f(x^k) - d^k\right\|_2^2 \\
&= \left\|\nabla f(x^k) - \alpha_k\nabla f(x^k, \xi_k) - (1 - \alpha_k)d^{k-1}\right\|_2^2 \\
&= \left\|\alpha_k\left(\nabla f(x^k) - \nabla f(x^k, \xi_k)\right) + (1 - \alpha_k)\left(\nabla f(x^k) - \nabla f(x^{k-1})\right) - (1 - \alpha_k)\left(d^{k-1} - \nabla f(x^{k-1})\right)\right\|_2^2 \\
&= \alpha_k^2\left\|\nabla f(x^k) - \nabla f(x^k, \xi_k)\right\|_2^2 + (1 - \alpha_k)^2\left\|\nabla f(x^k) - \nabla f(x^{k-1})\right\|_2^2 \\
&\quad + (1 - \alpha_k)^2\left\|\nabla f(x^{k-1}) - d^{k-1}\right\|_2^2 \\
&\quad + 2\alpha_k(1 - \alpha_k)\langle\nabla f(x^{k-1}) - \nabla f(x^{k-1}, \xi_{k-1}), \nabla f(x^k) - \nabla f(x^{k-1})\rangle \\
&\quad + 2\alpha_k(1 - \alpha_k)\langle\nabla f(x^k) - \nabla f(x^k, \xi_k), \nabla f(x^{k-1}) - d^{k-1}\rangle \\
&\quad + 2(1 - \alpha_k)^2\langle\nabla f(x^k) - \nabla f(x^{k-1}), \nabla f(x^{k-1}) - d^{k-1}\rangle.
\end{aligned}
$$

Taking the expectation conditioned on the filtration $\mathcal{F}_k$ generated by the iterates until $k$, i.e., the sigma algebra generated by $\{x_1, \ldots, x_k\}$, which we denote using $\mathbb{E}_k[\cdot]$, and using the unbiased property in Assumption 5.3, we get,

$$\mathbb{E}_k[\|\lambda^k\|_2^2] = \alpha_k^2 \mathbb{E}_k[\|\nabla f(x^k) - \nabla f(x^k, \xi_k)\|_2^2] + (1 - \alpha_k)^2 \|\nabla f(x^k) - \nabla f(x^{k-1})\|_2^2$$
$$+ (1 - \alpha_k)^2 \|\lambda^{k-1}\|_2^2 + 2(1 - \alpha_k)^2 \langle \nabla f(x^k) - \nabla f(x^{k-1}), \lambda^{k-1} \rangle.$$

From this expression we can estimate,

$$\mathbb{E}_k[\|\lambda^k\|_2^2] \overset{(a)}{\leq} \alpha_k^2 \sigma^2 + (1 - \alpha_k)^2 \|\nabla f(x^k) - \nabla f(x^{k-1})\|_2^2 + (1 - \alpha_k)^2 \|\lambda^{k-1}\|_2^2 + 2(1 - \alpha_k)^2 \langle \nabla f(x^k) - \nabla f(x^{k-1}), \lambda^{k-1} \rangle$$

$$\overset{(b)}{\leq} \alpha_k^2 \sigma^2 + (1 - \alpha_k)^2 \|\nabla f(x^k) - \nabla f(x^{k-1})\|_2^2 + (1 - \alpha_k)^2 \|\lambda^{k-1}\|_2^2$$
$$+ (1 - \alpha_k)^2 \left( \tfrac{\alpha_k}{2} \|\nabla f(x^k) - \nabla f(x^{k-1})\|_2^2 + \tfrac{2}{\alpha_k} \|\lambda^{k-1}\|_2^2 \right)$$

$$\overset{(c)}{\leq} \alpha_k^2 \sigma^2 + (1 - \alpha_k)^2 L_2^2 \|x^k - x^{k-1}\|_2^2 + (1 - \alpha_k)^2 \|\lambda^{k-1}\|_2^2 + (1 - \alpha_k)^2 \left( (\tfrac{\alpha_k}{2}) L_2^2 \|x^k - x^{k-1}\|_2^2 + \tfrac{2}{\alpha_k} \|\lambda^{k-1}\|_2^2 \right)$$

$$\overset{(d)}{\leq} \alpha_k^2 \sigma^2 + (1 - \alpha_k)^2 L_2^2 \rho_2^2 \gamma^2 + (1 - \alpha_k)^2 \|\lambda^{k-1}\|_2^2 + (1 - \alpha_k)^2 \left( (\tfrac{\alpha_k}{2}) L_2^2 \rho_2^2 \gamma^2 + \tfrac{2}{\alpha_k} \|\lambda^{k-1}\|_2^2 \right)$$

$$\overset{(e)}{\leq} \alpha_k^2 \sigma^2 + (1 + \tfrac{\alpha_k}{2})(1 - \alpha_k) L_2^2 \rho_2^2 \gamma^2 + (1 + \tfrac{2}{\alpha_k})(1 - \alpha_k) \|\lambda^{k-1}\|_2^2,$$

using the bounded variance property from Assumption 5.3 for (a), Young's inequality with parameter $\alpha_k/2 > 0$ for (b), the Lipschitz property of $f$ under norm $\|\cdot\|_2$ for (c), the update definition from Algorithm 1 for (d), and the fact that $1 - \alpha_k < 1$ for (e). To complete the proof, we note that

$$(1 + \tfrac{2}{\alpha_k})(1 - \alpha_k) \leq (1 - \tfrac{\alpha_k}{2}) \quad \text{and} \quad (1 - \alpha_k)(1 + \tfrac{\alpha_k}{2}) \leq \tfrac{2}{\alpha_k}$$

which, applied to the previous inequality and taking total expectations, yields

$$\mathbb{E}[\|\lambda^k\|_2^2] \leq \left( 1 - \frac{\alpha_k}{2} \right) \mathbb{E}[\|\lambda^{k-1}\|_2^2] + \alpha_k^2 \sigma^2 + \frac{2 L_2^2 \rho_2^2 \gamma^2}{\alpha_k}.$$

$\square$

### D.1.1. CONSTANT $\alpha$

**Lemma D.3.** *Suppose Assumptions 5.1 and 5.3 hold. Let $n \in \mathbb{N}^*$ and consider the iterates $\{x^k\}_{k=1}^n$ generated by Algorithm 1 with constant stepsize $\gamma > 0$ and constant momentum $\alpha \in (0, 1)$ with the exception of the first iteration, where we take $\alpha = 1$. Then, we have for all $k \in \{1, \ldots, n\}$*

$$\sqrt{\mathbb{E}[\|\lambda^k\|_2^2]} \leq \frac{\sqrt{2} L_2 \rho_2 \gamma}{\alpha} + \left( \sqrt{\alpha} + \left( \sqrt{1 - \frac{\alpha}{2}} \right)^k \right) \sigma.$$

*Proof.* Let $n \in \mathbb{N}^*$, $k \in \{1, \ldots, n\}$, and invoke Lemma D.2 to get

$$\mathbb{E}[\|\lambda^k\|_2^2] \leq \left( 1 - \frac{\alpha}{2} \right) \mathbb{E}[\|\lambda^{k-1}\|_2^2] + \frac{2 L_2^2 \rho_2^2 \gamma^2}{\alpha} + \alpha^2 \sigma^2.$$

Applying Lemma D.9 with $\beta = \frac{\alpha}{2}$ and $\eta = \frac{2 L_2^2 \rho_2^2 \gamma^2}{\alpha} + \alpha^2 \sigma^2$ gives directly

$$\mathbb{E}[\|\lambda^k\|_2^2] \leq \frac{2 L_2^2 \rho_2^2 \gamma^2}{\alpha^2} + \alpha \sigma^2 + \left( 1 - \frac{\alpha}{2} \right)^k \mathbb{E}[\|\lambda^1\|_2^2]$$
$$\leq \frac{2 L_2^2 \rho_2^2 \gamma^2}{\alpha^2} + \left( \alpha + \left( 1 - \frac{\alpha}{2} \right)^k \right) \sigma^2$$

after using Assumption 5.3 in the final inequality. Taking square roots and upper bounding then yields

$$\sqrt{\mathbb{E}[\|\lambda^k\|_2^2]} \leq \frac{\sqrt{2} L_2 \rho_2 \gamma}{\alpha} + \left( \sqrt{\alpha} + \left( \sqrt{1 - \frac{\alpha}{2}} \right)^k \right) \sigma.$$

$\square$

**Theorem 5.4** (Convergence rate for uSCG with constant $\alpha$). *Suppose Assumptions 5.1 and 5.3 hold. Let $n \in \mathbb{N}^*$ and consider the iterates $\{x^k\}_{k=1}^n$ generated by Algorithm 1 with constant stepsize $\gamma = \frac{1}{\sqrt{n}}$ and constant momentum $\alpha \in (0,1)$. Then, it holds that*

$$\mathbb{E}[\|\nabla f(\bar{x}^n)\|_*] \leq O\left(\frac{L\rho}{\sqrt{n}} + \sigma\right).$$

*Proof.* Let $n \in \mathbb{N}^*$; we will first invoke Lemma D.1 and then we will estimate the error terms inside using Lemma D.2 under Assumptions 5.1 and 5.3. As shown in Lemma D.1,

$$\mathbb{E}[\|\nabla f(\bar{x}^n)\|_*] \leq \frac{\mathbb{E}[f(x^1) - f^\star]}{\rho\gamma n} + \frac{L\rho\gamma}{2n} + \frac{1}{n}\left(\frac{\rho_2}{\rho} + \zeta\right)\sum_{k=1}^n \sqrt{\mathbb{E}[\|\lambda^k\|_2^2]}. \tag{13}$$

By Lemma D.2 with Lemma D.9, we get

$$\sqrt{\mathbb{E}[\|\lambda^k\|_2^2]} \leq \frac{\sqrt{2}L_2\rho_2\gamma}{\alpha} + \left(\sqrt{\alpha} + \left(\sqrt{1 - \frac{\alpha}{2}}\right)^k\right)\sigma$$

which, if we sum from $k = 1$ to $n$, gives us

$$\sum_{k=1}^n \sqrt{\mathbb{E}[\|\lambda^k\|_2^2]} \leq n\frac{\sqrt{2}L_2\rho_2\gamma}{\alpha} + \left(n\sqrt{\alpha} + \frac{\sqrt{1 - \frac{\alpha}{2}}}{1 - \sqrt{1 - \frac{\alpha}{2}}}\right)\sigma.$$

Plugging this estimate into Equation (13) gives

$$\begin{aligned}
\mathbb{E}[\|\nabla f(\bar{x}^n)\|_*] &\leq \frac{\mathbb{E}[f(x^1) - f^\star]}{\rho\gamma n} + \frac{L\rho\gamma}{2} + \frac{1}{n}\left(\frac{\rho_2}{\rho} + \zeta\right)\sum_{k=1}^n \mathbb{E}[\|\lambda^k\|_2] \\
&\leq \frac{\mathbb{E}[f(x^1) - f^\star]}{\rho\gamma n} + \frac{L\rho\gamma}{2} + \frac{1}{n}\left(\frac{\rho_2}{\rho} + \zeta\right)\left(n\frac{\sqrt{2}L_2\rho_2\gamma}{\alpha} + \left(n\sqrt{\alpha} + \frac{\sqrt{1 - \frac{\alpha}{2}}}{1 - \sqrt{1 - \frac{\alpha}{2}}}\right)\sigma\right) \\
&= \frac{\mathbb{E}[f(x^1) - f^\star]}{\rho\gamma n} + \frac{L\rho\gamma}{2} + \left(\frac{\rho_2}{\rho} + \zeta\right)\left(\frac{\sqrt{2}L_2\rho_2\gamma}{\alpha} + \left(\sqrt{\alpha} + \frac{\sqrt{1 - \frac{\alpha}{2}}}{n(1 - \sqrt{1 - \frac{\alpha}{2}})}\right)\sigma\right).
\end{aligned} \tag{14}$$

Finally, by substituting $\gamma = \frac{1}{\sqrt{n}}$ and noting $f(x^{n+1}) \geq f^\star$ we arrive at

$$\begin{aligned}
\mathbb{E}[\|\nabla f(\bar{x}^n)\|_*] &\leq \frac{\mathbb{E}[f(x^1) - f^\star]}{\sqrt{n}\rho} + \frac{L\rho}{2\sqrt{n}} + \left(\frac{\rho_2}{\rho} + \zeta\right)\left(\frac{\sqrt{2}L_2\rho_2}{\alpha\sqrt{n}} + \left(\sqrt{\alpha} + \frac{\sqrt{1 - \frac{\alpha}{2}}}{n(1 - \sqrt{1 - \frac{\alpha}{2}})}\right)\sigma\right) \\
&= O\left(\frac{1}{\sqrt{n}} + \sigma\right).
\end{aligned}$$

$\square$

### D.1.2. VANISHING $\alpha_k$

**Lemma D.4** (Bound on the gradient error with vanishing $\alpha$). *Suppose Assumptions 5.1 and 5.3 hold. Let $n \in \mathbb{N}^*$ and consider the iterates $\{x_k\}_{k=1}^n$ generated by Algorithm 1 with a constant stepsize $\gamma$ satisfying*

$$\frac{1}{2n^{3/4}} < \gamma < \frac{1}{n^{3/4}}. \tag{15}$$

*Moreover, consider momentum which vanishes $\alpha_k = \frac{1}{\sqrt{k}}$. Then, for all $k \in \{1, \ldots, n\}$ the following holds*

$$\mathbb{E}[\|\lambda^k\|_2^2] \leq \frac{4\sigma^2 + 8L_2^2\rho_2^2}{\sqrt{k}}. \tag{16}$$

*Proof.* Let $k \in \{1, \ldots, n\}$, then by invoking the recursive inequality obtained in Lemma D.2 for $\mathbb{E}[\|\lambda^k\|_2^2]$ we have,

$$\mathbb{E}[\|\lambda^k\|_2^2] \leq \left(1 - \frac{\alpha_k}{2}\right)\mathbb{E}[\|\lambda^{k-1}\|_2^2] + \alpha_k^2\sigma^2 + \frac{2L_2^2\rho_2^2\gamma^2}{\alpha_k}. \tag{17}$$

Using the particular choice of $\gamma$ given in the statement of the lemma,

$$\frac{1}{2n^{3/4}} < \gamma < \frac{1}{n^{3/4}}, \tag{18}$$

as well as the choice of $\alpha_k$ and the fact that $n \geq k$, we get

$$\begin{aligned}
\mathbb{E}[\|\lambda^k\|_2^2] &\leq \left(1 - \frac{\alpha_k}{2}\right)\mathbb{E}[\|\lambda^{k-1}\|_2^2] + \alpha_k^2\sigma^2 + \frac{2L_2^2\rho_2^2}{\alpha_k n^{3/2}} \\
&\leq \left(1 - \frac{\alpha_k}{2}\right)\mathbb{E}[\|\lambda^{k-1}\|_2^2] + \alpha_k^2\sigma^2 + \frac{2L_2^2\rho_2^2}{\alpha_k k^{3/2}} \\
&= \left(1 - \frac{1}{2\sqrt{k}}\right)\mathbb{E}[\|\lambda^{k-1}\|_2^2] + \frac{\sigma^2}{k} + \frac{2L_2^2\rho_2^2}{k} \\
&= \left(1 - \frac{1}{2\sqrt{k}}\right)\mathbb{E}[\|\lambda^{k-1}\|_2^2] + \frac{\sigma^2 + 2L_2^2\rho_2^2}{k}.
\end{aligned}$$

Then, by applying Lemma D.10 with $u^k = \mathbb{E}[\|\lambda^k\|_2^2]$ and $c = \sigma^2 + 2L_2^2\rho_2^2$ we readily obtain

$$\mathbb{E}[\|\lambda^k\|_2^2] \leq \frac{4\sigma^2 + 8L_2^2\rho_2^2}{\sqrt{k}} \tag{19}$$

since $Q$ as defined in Lemma D.10 is given by $Q = \max\{\mathbb{E}[\|\lambda^1\|_2^2], 4\sigma^2 + 8L_2^2\rho_2^2\} \leq 4\sigma^2 + 8L_2^2\rho_2^2$, which concludes our result. $\qquad\square$

Combining these results yields our accuracy guarantees for Algorithm 1 with vanishing $\alpha_k$, presented in the next lemma.

**Theorem 5.5** (Convergence rate for uSCG with vanishing $\alpha_k$)**.** *Suppose that Assumptions 5.1 and 5.3 hold. Let $n \in \mathbb{N}^*$ and consider the iterates $\{x^k\}_{k=1}^n$ generated by Algorithm 1 with a constant stepsize $\gamma$ satisfying $\frac{1}{2n^{3/4}} < \gamma < \frac{1}{n^{3/4}}$ and vanishing momentum $\alpha_k = \frac{1}{\sqrt{k}}$. Then, it holds that*

$$\mathbb{E}[\|\nabla f(\bar{x}^n)\|_*] = O\left(\frac{1}{n^{1/4}} + \frac{L\rho}{n^{3/4}}\right).$$

*Proof.* Let $n \in \mathbb{N}^*$, $k \in \{1, \ldots, n\}$; by combining Lemma D.1 and Lemma D.4 we have

$$\begin{aligned}
\mathbb{E}[\|\nabla f(\bar{x}^n)\|_*] &\overset{\text{(D.1)}}{\leq} \frac{2\mathbb{E}[f(x^1) - f^\star]}{\rho n^{1/4}} + \frac{2(\rho_2 + \zeta\rho)\sum_{k=1}^n \sqrt{\mathbb{E}[\|\lambda^k\|_2^2]}}{\rho n} + \frac{L\rho}{n^{3/4}} \\
&\overset{\text{(D.4)}}{\leq} \frac{2\mathbb{E}[f(x^1) - f^\star]}{\rho n^{1/4}} + \frac{2(\rho_2 + \zeta\rho)\sqrt{4\sigma^2 + 8L_2^2\rho_2^2}\sum_{k=1}^n \frac{1}{k^{1/4}}}{\rho n} + \frac{L\rho}{n^{3/4}} \\
&\leq \frac{2\mathbb{E}[f(x^1) - f^\star]}{\rho n^{1/4}} + \frac{2(\rho_2 + \zeta\rho)\sqrt{4\sigma^2 + 8L_2^2\rho_2^2}\sum_{k=1}^n \frac{1}{k^{1/4}}}{\rho n} + \frac{L\rho}{n^{3/4}}.
\end{aligned} \tag{20}$$

Using the integral test and noting that $x \mapsto \frac{1}{x^{1/4}}$ is decreasing on $\mathbb{R}_+$, we can upper bound the sum in the right hand side as

$$\sum_{k=1}^n \frac{1}{k^{1/4}} \leq 1 + \int_1^n \frac{1}{x^{3/4}}dx = 1 + \frac{4}{3}[x^{3/4}]_1^n = 1 + \frac{4}{3}(n^{3/4} - 1) = \frac{4}{3}n^{3/4} - \frac{1}{3} \leq \frac{4}{3}n^{3/4}.$$

Inserting the above estimation into (20) we arrive at

$$\mathbb{E}[\|\nabla f(\bar{x}^n)\|_*] \leq \frac{2\mathbb{E}[f(x^1) - f^\star]}{\rho n^{1/4}} + \frac{8n^{3/4}(\rho_2 + \zeta\rho)\sqrt{4\sigma^2 + 8L_2^2\rho_2^2}}{3\rho n} + \frac{L\rho}{n^{3/4}}$$

$$= \frac{2\mathbb{E}[f(x^1) - f^\star] + \frac{8}{3}(\rho_2 + \zeta\rho)\sqrt{4\sigma^2 + 8L_2^2\rho_2^2}}{\rho n^{1/4}} + \frac{L\rho}{n^{3/4}}$$

$$= O\left(\frac{1}{n^{1/4}} + \frac{L\rho}{n^{3/4}}\right)$$

which is the claimed result. $\qquad\square$

## D.2. Convergence analysis of SCG

In this section we will analyze the worst-case convergence rate of Algorithm 2. To do this, we will prove bounds on the expectation of the so-called Frank-Wolfe gap, $\max_{u \in \mathcal{D}}\langle \nabla f(x), x - u\rangle$, which ensures criticality for the constrained optimization problem over $\mathcal{D}$, i.e., for $x^\star \in \mathcal{D}$

$$0 = \nabla f(x^\star) + \mathrm{N}_{\mathcal{D}}(x^\star) \iff \max_{u \in \mathcal{D}}\langle \nabla f(x^\star), x^\star - u\rangle \leq 0$$

where $\mathrm{N}_{\mathcal{D}}$ is the normal cone to the set convex $\mathcal{D}$.

This next lemma characterizes the descent of Algorithm 2 for any stepsize $\gamma$ and momentum $\alpha_k$ in $(0, 1]$.

**Lemma D.5** (Nonconvex analog Mokhtari et al. (2020, Lem. 2)). *Suppose Assumption 5.1 holds. Let $n \in \mathbb{N}^*$ and consider the iterates $\{x_k\}_{k=1^n}$ generated by Algorithm 2 with constant stepsize $\gamma \in (0, 1]$. Then, for all $k \in \{1, \ldots, n\}$, for all $u \in \mathcal{D}$, it holds*

$$\gamma\mathbb{E}[\langle \nabla f(x^k), x^k - u\rangle] \leq \mathbb{E}[f(x^k) - f(x^{k+1})] + D_2\gamma\sqrt{\mathbb{E}[\|\lambda^k\|_2^2]} + 2L\rho^2\gamma^2. \tag{21}$$

*Proof.* Let $n \in \mathbb{N}^*$, then by Assumption 5.1 we can apply the descent lemma for the function $f$ at the points $x^k$ and $x^{k+1}$ to get, for all $k \in \{1, \ldots, n\}$,

$$f(x^{k+1}) \leq f(x^k) + \langle \nabla f(x^k), x^{k+1} - x^k\rangle + \tfrac{L}{2}\|x^{k+1} - x^k\|^2$$

$$= f(x^k) + \langle d^k, x^{k+1} - x^k\rangle + \langle \lambda^k, x^{k+1} - x^k\rangle + \tfrac{L}{2}\|x^{k+1} - x^k\|^2$$

$$= f(x^k) + \gamma\langle d^k, \mathrm{lmo}(d^k) - x^k\rangle + \gamma\langle \lambda^k, \mathrm{lmo}(d^k) - x^k\rangle + \tfrac{L}{2}\gamma^2\|\mathrm{lmo}(d^k) - x^k\|^2$$

$$\overset{(a)}{\leq} f(x^k) + \gamma\langle d^k, u - x^k\rangle + \gamma\langle \lambda^k, \mathrm{lmo}(d^k) - x^k\rangle + \tfrac{L}{2}\gamma^2\|\mathrm{lmo}(d^k) - x^k\|^2$$

$$= f(x^k) + \gamma\langle -\lambda^k, u - x^k\rangle + \gamma\langle \nabla f(x^k), u - x^k\rangle + \gamma\langle \lambda^k, \mathrm{lmo}(d^k) - x^k\rangle + \tfrac{L}{2}\gamma^2\|\mathrm{lmo}(d^k) - x^k\|^2$$

$$= f(x^k) + \gamma\langle \nabla f(x^k), u - x^k\rangle + \gamma\langle \lambda^k, \mathrm{lmo}(d^k) - u\rangle + \tfrac{L}{2}\gamma^2\|\mathrm{lmo}(d^k) - x^k\|^2$$

$$\overset{(b)}{\leq} f(x^k) + \gamma\langle \nabla f(x^k), u - x^k\rangle + \gamma\langle \lambda^k, \mathrm{lmo}(d^k) - u\rangle + 2L\rho^2\gamma^2,$$

using the optimality of $\mathrm{lmo}(d^k)$ for the linear minimization subproblem for (a) and the $2\rho$ upper bound on $\|\mathrm{lmo}(d^k) - x^k\|$ for (b). Rearranging and estimating we find, for all $k \in \{1, \ldots, n\}$, for all $u \in \mathcal{D}$,

$$\gamma\langle \nabla f(x^k), x^k - u\rangle \overset{(a)}{\leq} f(x^k) - f(x^{k+1}) + \gamma\|\lambda^k\|_2\|\mathrm{lmo}(d^k) - u\|_2 + \tfrac{L}{2}\gamma^2\|\mathrm{lmo}(d^k) - x^k\|^2$$

$$\overset{(b)}{\leq} f(x^k) - f(x^{k+1}) + D_2\gamma\|\lambda^k\|_2 + 2L\rho^2\gamma^2$$

where we have used the Cauchy-Schwarz inequality in (a) and and bounded $\|\mathrm{lmo}(d^k) - x^k\|_2$ using the diameter of the set $\mathcal{D}$ with respect to the Euclidean norm, denoted $D_2$, in (b). Taking the expectation of both sides and applying Jensen's inequality we finally arrive, for all $k \in \{1, \ldots, n\}$, for all $u \in \mathcal{D}$,

$$\gamma\mathbb{E}[\langle \nabla f(x^k), x^k - u\rangle] \leq \mathbb{E}[f(x^k) - f(x^{k+1})] + D_2\gamma\mathbb{E}[\|\lambda^k\|_2] + 2L\rho^2\gamma^2$$

$$\leq \mathbb{E}[f(x^k) - f(x^{k+1})] + D_2\gamma\sqrt{\mathbb{E}[\|\lambda^k\|_2^2]} + 2L\rho^2\gamma^2.$$

$\square$

D.2.1. SCG WITH CONSTANT $\alpha$

**Lemma D.6.** *Suppose Assumptions 5.1 and 5.3 hold. Let $n \in \mathbb{N}^*$ and consider the iterates $\{x^k\}_{k=1}^n$ generated by Algorithm 2 with constant stepsize $\gamma = \frac{1}{\sqrt{n}}$ and constant momentum $\alpha \in (0,1)$ with the exception of the first iteration, where we take $\alpha = 1$. Then we have*

$$\mathbb{E}[\|\lambda^k\|_2^2] \leq 4L_2^2 D_2^2 \frac{\gamma^2}{\alpha^2} + \left(2\alpha + \left(1 - \frac{\alpha}{2}\right)^k\right)\sigma^2.$$

*Proof.* Under Assumptions 5.1 and 5.3, Lemma 1 in Mokhtari et al. (2020) yields, after taking expectations, for all $k \in \{1, \ldots, n\}$

$$\mathbb{E}[\|\lambda^{k+1}\|_2^2] \leq (1 - \frac{\alpha_{k+1}}{2})\mathbb{E}[\|\lambda^k\|_2^2] + \sigma^2 \alpha_{k+1}^2 + 2L_2^2 D_2^2 \frac{\gamma^2}{\alpha_{k+1}}.$$

Taking $\gamma$ and $\alpha$ to be constant we get

$$\mathbb{E}[\|\lambda^{k+1}\|_2^2] \leq (1 - \frac{\alpha}{2})\mathbb{E}[\|\lambda^k\|_2^2] + \sigma^2 \alpha^2 + 2L_2^2 D_2^2 \frac{\gamma^2}{\alpha}.$$

Applying Lemma D.9 to the above with $u^k = \mathbb{E}[\|\lambda^{k+1}\|_2^2]$, $\beta = \frac{\alpha}{2}$, and $\eta = \sigma^2 \alpha^2 + 2L_2^2 D_2^2 \frac{\gamma^2}{\alpha}$ we obtain

$$\mathbb{E}[\|\lambda^k\|_2^2] \leq 2\alpha\sigma^2 + 4L_2^2 D_2^2 \frac{\gamma^2}{\alpha^2} + \left(1 - \frac{\alpha}{2}\right)^k \mathbb{E}[\|\lambda^1\|_2^2]$$

$$\leq 4L_2^2 D_2^2 \frac{\gamma^2}{\alpha^2} + \left(2\alpha + \left(1 - \frac{\alpha}{2}\right)^k\right)\sigma^2$$

with the final inequality following by the variance bound in Assumption 5.3. $\square$

**Theorem 5.6** (Convergence rate for SCG with constant $\alpha$). *Suppose Assumptions 5.1 and 5.3 hold. Let $n \in \mathbb{N}^*$ and consider the iterates $\{x^k\}_{k=1}^n$ generated by Algorithm 2 with constant stepsize $\gamma = \frac{1}{\sqrt{n}}$ and constant momentum $\alpha \in (0,1)$. Then, for all $u \in \mathcal{D}$, it holds that*

$$\mathbb{E}[\langle \nabla f(\bar{x}^n), \bar{x}^n - u \rangle] = O\left(\frac{L\rho^2}{\sqrt{n}} + \sigma\right).$$

*Proof.* Let $n \in \mathbb{N}^*$ and let $k \in \{1, \ldots, n\}$. By Assumption 5.1, we can invoke Lemma D.5 to get, for all $k \in \{1, \ldots, n\}$, for all $u \in \mathcal{D}$,

$$\gamma\mathbb{E}[\langle \nabla f(x^k), x^k - u \rangle] \leq \mathbb{E}[f(x^k) - f(x^{k+1})] + D_2\gamma\sqrt{\mathbb{E}[\|\lambda^k\|_2^2]} + 2L\rho^2\gamma^2.$$

Since Assumption 5.3 holds, we can then invoke Lemma D.6 and apply this to the above. This gives, for all $u \in \mathcal{D}$

$$\gamma\mathbb{E}[\langle \nabla f(x^k), x^k - u \rangle] \leq \mathbb{E}[f(x^k) - f(x^{k+1})] + 2L\rho^2\gamma^2 + D_2\gamma\sqrt{4L_2^2 D_2^2 \frac{\gamma^2}{\alpha^2} + \left(2\alpha + \left(1 - \frac{\alpha}{2}\right)^k\right)\sigma^2}$$

$$\leq \mathbb{E}[f(x^k) - f(x^{k+1})] + 2L\rho^2\gamma^2 + 2L_2 D_2^2 \frac{\gamma^2}{\alpha} + D_2\gamma\left(\sqrt{2\alpha} + \left(\sqrt{1 - \frac{\alpha}{2}}\right)^k\right)\sigma.$$

Summing from $k = 1$ to $n$ then dividing by $n\gamma$ we find, for all $u \in \mathcal{D}$,

$$\mathbb{E}[\langle \nabla f(\bar{x}^n), \bar{x}^n - u \rangle] = \frac{1}{n}\sum_{k=1}^n \mathbb{E}[\langle \nabla f(x^k), x^k - u \rangle]$$

$$\overset{(a)}{\leq} \frac{\mathbb{E}[f(x^1) - f(x^{n+1})]}{\gamma n} + 2L\rho^2\gamma + 2L_2 D_2^2 \frac{\gamma}{\alpha} + D_2\left(\sqrt{2\alpha} + \frac{1}{n}\sum_{k=1}^n \left(\sqrt{1 - \frac{\alpha}{2}}\right)^k\right)\sigma$$

$$\overset{(b)}{\leq} \frac{\mathbb{E}[f(x^1) - f(x^{n+1})]}{\gamma n} + 2L\rho^2\gamma + 2L_2 D_2^2 \frac{\gamma}{\alpha} + D_2\left(\sqrt{2\alpha} + \frac{\sqrt{1 - \frac{\alpha}{2}}}{n\left(1 - \sqrt{1 - \frac{\alpha}{2}}\right)}\right)\sigma$$

$$\overset{(c)}{\leq} \frac{\mathbb{E}[f(x^1) - f^\star]}{\gamma n} + 2L\rho^2\gamma + 2L_2 D_2^2 \frac{\gamma}{\alpha} + D_2\left(\sqrt{2\alpha} + \frac{\sqrt{1 - \frac{\alpha}{2}}}{n\left(1 - \sqrt{1 - \frac{\alpha}{2}}\right)}\right)\sigma,$$

$(22)$

applying the subadditivity of the square root for (a), geometric series due to $\sqrt{1 - \frac{\alpha}{2}} \in (0, 1)$ for (b), and the definition of $f^\star$ for (c). Taking $\gamma = \frac{1}{\sqrt{n}}$ then gives the final result, for all $u \in \mathcal{D}$,

$$\mathbb{E}[\langle \nabla f(\bar{x}^n), \bar{x}^n - u \rangle] \leq \frac{\mathbb{E}[f(x^1) - f^\star]}{\sqrt{n}} + \frac{2L\rho^2}{\sqrt{n}} + \frac{2L_2 D_2^2}{\alpha \sqrt{n}} + D_2 \left( \sqrt{2\alpha} + \frac{\sqrt{1 - \frac{\alpha}{2}}}{n \left( 1 - \sqrt{1 - \frac{\alpha}{2}} \right)} \right) \sigma \quad = O \left( \frac{L\rho^2}{\sqrt{n}} + \sigma \right).$$

$\square$

### D.2.2. SCG WITH VANISHING $\alpha$

We now proceed to analyze the convergence of Algorithm 2 with vanishing $\alpha_k$. The next lemma provides an estimation on the decay of the second moment of the noise $\lambda^k$.

**Lemma D.7** (Bound on the gradient error with vanishing $\alpha$ Algorithm 2). *Suppose Assumptions 5.1 and 5.3 hold. Let $n \in \mathbb{N}^*$ and consider the iterates $\{x_k\}_{k=1}^n$ generated by Algorithm 2 with a constant stepsize $\gamma$ satisfying*

$$\frac{1}{2n^{3/4}} < \gamma < \frac{1}{n^{3/4}}. \tag{23}$$

*Moreover, consider vanishing momentum $\alpha_k = \frac{1}{\sqrt{k}}$. Then, for all $k \in \{1, \ldots, n\}$ the following holds*

$$\mathbb{E}[\|\lambda^k\|_2^2] \leq \frac{4\sigma^2 + 8L_2^2 D_2^2}{\sqrt{k}}. \tag{24}$$

*Proof.* Under Assumptions 5.1 and 5.3, we have the following recursion from Lemma 1 in Mokhtari et al. (2020) after taking expectations, for all $k \in \mathbb{N}^*$,

$$\mathbb{E}[\|\lambda^{k+1}\|_2^2] \leq (1 - \frac{\alpha_{k+1}}{2})\mathbb{E}[\|\lambda^k\|_2^2] + \sigma^2 \alpha_{k+1}^2 + 2L_2^2 D_2^2 \frac{\gamma^2}{\alpha_{k+1}}.$$

Comparing with the bound in Lemma D.4, we see the only difference is the change of the constant $D_2^2$ by $\rho_2^2$. Repeating the argument in Lemma D.4, the desired claim is directly obtained with $D_2^2$ in place of $\rho_2^2$, with the constant $Q = \max\{\mathbb{E}[\|\lambda^1\|_2^2], 4\sigma^2 + 8L_2^2 D_2^2\} \leq 4\sigma^2 + 8L_2^2 D_2^2$ since $\mathcal{E}[\|\lambda^1\|_2^2] \leq \sigma^2$ by Assumption 5.3. $\square$

**Theorem 5.7** (Convergence rate for SCG with vanishing $\alpha_k$). *Suppose Assumptions 5.1 and 5.3 hold. Let $n \in \mathbb{N}^*$ and consider the iterates $\{x^k\}_{k=1}^n$ generated by Algorithm 2 with a constant stepsize $\gamma$ satisfying $\frac{1}{2n^{3/4}} < \gamma < \frac{1}{n^{3/4}}$ and vanishing momentum $\alpha_k = \frac{1}{\sqrt{k}}$. Then, for all $u \in \mathcal{D}$, it holds that*

$$\mathbb{E}[\langle \nabla f(\bar{x}^n), \bar{x}^n - u \rangle] = O \left( \frac{1}{n^{1/4}} + \frac{L\rho^2}{n^{3/4}} \right).$$

*Proof.* Let $n \in \mathbb{N}^*$ and $k \in \{1, \ldots, n\}$. By Assumption 5.1, we can invoke Lemma D.5 to get,

$$\gamma \mathbb{E}[\langle \nabla f(x^k), x^k - u \rangle] \leq \mathbb{E}[f(x^k) - f(x^{k+1})] + D_2 \gamma \sqrt{\mathbb{E}[\|\lambda^k\|_2^2]} + 2L\rho^2 \gamma^2.$$

Applying the estimate given in Lemma D.7 to the above we get

$$\gamma \mathbb{E}[\langle \nabla f(x^k), x^k - u \rangle] \leq \mathbb{E}[f(x^k) - f(x^{k+1})] + D_2 \gamma \sqrt{\frac{4\sigma^2 + 8L_2^2 D_2^2}{\sqrt{k}}} + 2L\rho^2 \gamma^2$$

$$= \mathbb{E}[f(x^k) - f(x^{k+1})] + D_2 \sqrt{4\sigma^2 + 8L_2^2 D_2^2} \gamma \frac{1}{k^{1/4}} + 2L\rho^2 \gamma^2.$$

Summing from $k = 1$ to $n$ and then dividing by $n\gamma$ we find, for all $u \in \mathcal{D}$,

$$
\begin{aligned}
\mathbb{E}[\langle \nabla f(\bar{x}^n), \bar{x}^n - u\rangle] &= \frac{1}{n}\sum_{k=1}^{n}\mathbb{E}[\langle \nabla f(x^k), x^k - u\rangle] \\
&\overset{(a)}{\leq} \frac{\mathbb{E}[f(x^1) - f(x^{n+1})]}{n\gamma} + \frac{D_2\sqrt{4\sigma^2 + 8L_2^2 D_2^2}}{n}\sum_{k=1}^{n}\frac{1}{k^{1/4}} + 2L\rho^2\gamma \\
&\overset{(b)}{\leq} \frac{\mathbb{E}[f(x^1) - f(x^{n+1})]}{n\gamma} + \frac{4D_2\sqrt{4\sigma^2 + 8L_2^2 D_2^2}n^{3/4}}{3n} + 2L\rho^2\gamma \\
&= \frac{\mathbb{E}[f(x^1) - f(x^{n+1})]}{n\gamma} + \frac{4D_2\sqrt{4\sigma^2 + 8L_2^2 D_2^2}}{3n^{1/4}} + 2L\rho^2\gamma,
\end{aligned}
$$

using division by $\gamma n$ for (a) and the integral test with decreasing function $x \mapsto \frac{1}{x^{1/4}}$ for (b). Using the definition of $f^\star$ and estimating $n\gamma > \frac{n^{1/4}}{2}$ and $\gamma < \frac{1}{n^{3/4}}$ gives

$$
\begin{aligned}
\mathbb{E}[\langle \nabla f(\bar{x}^n), \bar{x}^n - u\rangle] &\leq \frac{2\mathbb{E}[f(x^1) - f^\star]}{n^{1/4}} + \frac{4D_2\sqrt{4\sigma^2 + 8L_2^2 D_2^2}}{3n^{1/4}} + \frac{2L\rho^2}{n^{3/4}} \\
&= O\left(\frac{1}{n^{1/4}} + \frac{L\rho^2}{n^{3/4}}\right).
\end{aligned}
$$

$\square$

### D.3. Averaged LMO Directional Descent (ALMOND)

In this section we present a variation on Algorithm 1 that computes the lmo directly on the stochastic gradient oracle and then does averaging. This is in contrast to how we have presented Algorithm 1 which first does averaging (aka momentum) with the stochastic gradient oracle and then computes the lmo. A special case of this algorithm is the Normalized SGD based algorithm of Zhao et al. (2020) when the set $\mathcal{D}$ is with respect to the Euclidean norm. In contrast with Algorithm 1, the method relies on large batches, since the noise is not controlled by the momentum parameter $\alpha$ due to the bias introduced by the lmo.

---

**Algorithm 4** Averaged LMO directioNal Descent (ALMOND)

---

**Input:** Horizon $n$, initialization $x^1 \in \mathcal{X}$, $d^0 = 0$, momentum $\alpha \in (0, 1)$, stepsize $\gamma \in (0, 1)$

1: **for** $k = 1, \ldots, n$ **do**
2:     Sample $\xi_k \sim \mathcal{P}$
3:     $d^k \leftarrow \alpha \, \mathrm{lmo}(\nabla f(x^k, \xi_k)) + (1 - \alpha)d^{k-1}$
4:     $x^{k+1} \leftarrow x^k + \gamma d^k$
5: Choose $\bar{x}^n$ uniformly at random from $\{x^1, \ldots, x^n\}$
**Return** $\bar{x}^n$

---

**Lemma D.8.** *Suppose Assumptions 5.1 and 5.3 hold. Let $n \in \mathbb{N}^*$ and consider the iterates $\{x_k\}_{k=1}^{n}$ generated by Algorithm 4 with stepsize $\gamma = \frac{1}{\sqrt{n}}$. Then, it holds*

$$
\mathbb{E}[\|\nabla f(\bar{x}^n)\|_*] \leq \frac{\mathbb{E}[f(x^1) - f^\star]}{\rho\sqrt{n}} + \frac{L(1 - \alpha)\rho}{\alpha\sqrt{n}} + \frac{L\rho}{2\sqrt{n}} + 2\mu\sigma = O\left(\frac{1}{\sqrt{n}}\right) + 2\mu\sigma
$$

*where[1] $\mu = \max\limits_{x \in \mathcal{X}} \frac{\|x\|_*}{\|x\|_2}$.*

*Proof.* Let $n \in \mathbb{N}^*$ and denote $z^k = \frac{1}{\alpha}x^k - \frac{1-\alpha}{\alpha}x^{k-1}$ with the convention that $x_0 = x_1$ so that $z_1 = x_1$ and, for all $k \in \{1, \ldots, n\}$,

$$
z^{k+1} - z^k = \frac{1}{\alpha}x^{k+1} - \frac{1-\alpha}{\alpha}x^k - \frac{1}{\alpha}x^k + \frac{1-\alpha}{\alpha}x^{k-1} = \frac{1}{\alpha}\left(\gamma d^k - \gamma(1-\alpha)d^{k-1}\right) = \gamma\,\mathrm{lmo}(g^k).
$$

---

[1]Alternatively, instead of invoking the constant $\mu$ we could make an assumption that the gradient oracle has bounded variance measured in the norm $\|\cdot\|_*$.

Applying the descent lemma for $f$ at the points $z^{k+1}$ and $z^k$ gives

$$
\begin{aligned}
f(z^{k+1}) &\leq f(z^k) + \langle \nabla f(z^k), z^{k+1} - z^k \rangle + \frac{L}{2} \left\| z^{k+1} - z^k \right\|^2 \\
&= f(z^k) + \gamma \langle \nabla f(z^k), \mathrm{lmo}(g^k) \rangle + \frac{L\gamma^2}{2} \left\| \mathrm{lmo}(g^k) \right\|^2 \\
&= f(z^k) + \gamma \left( \langle \nabla f(z^k) - \nabla f(x^k), \mathrm{lmo}(g^k) \rangle + \langle \nabla f(x^k) - g^k, \mathrm{lmo}(g^k) \rangle + \langle g^k, \mathrm{lmo}(g^k) \rangle \right) + \frac{L\gamma^2}{2} \left\| \mathrm{lmo}(g^k) \right\|^2 \\
&= f(z^k) + \gamma \left( \langle \nabla f(z^k) - \nabla f(x^k), \mathrm{lmo}(g^k) \rangle + \langle \nabla f(x^k) - g^k, \mathrm{lmo}(g^k) \rangle - \rho \left\| g^k \right\|_* \right) + \frac{L\gamma^2}{2} \left\| \mathrm{lmo}(g^k) \right\|^2 \\
&\overset{(a)}{\leq} f(z^k) + \gamma \left( \left( \left\| \nabla f(z^k) - \nabla f(x^k) \right\|_* + \left\| \nabla f(x^k) - g^k \right\|_* \right) \left\| \mathrm{lmo}(g^k) \right\| - \rho \left\| g^k \right\|_* \right) + \frac{L\gamma^2}{2} \left\| \mathrm{lmo}(g^k) \right\|^2 \\
&\overset{(b)}{\leq} f(z^k) + \gamma \left( \rho \left( \left\| \nabla f(z^k) - \nabla f(x^k) \right\|_* + \left\| \nabla f(x^k) - g^k \right\|_* \right) - \rho \left\| g^k \right\|_* \right) + \frac{L\rho^2\gamma^2}{2} \\
&\overset{(c)}{\leq} f(z^k) + \gamma \left( \rho \left( L \left\| z^k - x^k \right\| + \left\| \nabla f(x^k) - g^k \right\|_* \right) - \rho \left\| g^k \right\|_* \right) + \frac{L\rho^2\gamma^2}{2},
\end{aligned}
\tag{25}
$$

applying Hölder's inequality with norm $\left\| \cdot \right\|_*$ for (a), the radius $\rho$ of $\mathcal{D}$ for (b), and Assumption 5.1 for (c). We note that

$$
x^{k+1} - x^k = \gamma d^k = \gamma \left( (1-\alpha)d^{k-1} + \alpha \, \mathrm{lmo}(g^k) \right) = \alpha\gamma \, \mathrm{lmo}(g^k) + (1-\alpha)\gamma \left( \frac{x^k - x^{k-1}}{\gamma} \right) = \alpha\gamma \, \mathrm{lmo}(g^k) + (1-\alpha)(x^k - x^{k-1})
$$

which we can use to bound

$$
\left\| x^k - x^{k-1} \right\| \leq (1-\alpha) \left\| x^k - x^{k-1} \right\| + \alpha\gamma \left\| \mathrm{lmo}(g^k) \right\| \leq (1-\alpha) \left\| x^k - x^{k-1} \right\| + \alpha\rho\gamma \leq \frac{\alpha\rho\gamma}{(1-\alpha)}.
$$

We then have

$$
\left\| z^k - x^k \right\| = \frac{(1-\alpha)}{\alpha} \left\| x^k - x^{k-1} \right\| \leq \frac{(1-\alpha)\rho\gamma}{\alpha}
$$

by using the definition of the update and the $\mathrm{lmo}$, which can be plugged into (25) to get

$$
\begin{aligned}
\rho\gamma \left\| g^k \right\|_* &\leq f(z^k) - f(z^{k+1}) + \gamma\rho \left( L \left\| z^k - x^k \right\| + \left\| \nabla f(x^k) - g^k \right\|_* \right) + \frac{L\rho^2\gamma^2}{2} \\
&\overset{(a)}{\Longrightarrow} \left\| g^k \right\|_* \leq \frac{f(z^k) - f(z^{k+1})}{\rho\gamma} + L \left\| z^k - x^k \right\| + \left\| \nabla f(x^k) - g^k \right\|_* + \frac{L\rho\gamma}{2} \\
&\overset{(b)}{\leq} \frac{f(z^k) - f(z^{k+1})}{\rho\gamma} + \frac{L(1-\alpha)\rho\gamma}{\alpha} + \left\| \nabla f(x^k) - g^k \right\|_* + \frac{L\rho\gamma}{2} \\
&\overset{(c)}{\Longrightarrow} \left\| \nabla f(x^k) \right\|_* \leq \frac{(f(z^k) - f(z^{k+1}))}{\rho\gamma} + \frac{L(1-\alpha)\rho\gamma}{\alpha} + 2 \left\| \nabla f(x^k) - g^k \right\|_* + \frac{L\rho\gamma}{2}
\end{aligned}
\tag{26}
$$

where (a) is the result of dividing both sides by $\rho\gamma$, (b) is the result of bounding $\left\| z^k - x^k \right\|$, and (c) follows by the reverse triangle inequality after adding and subtracting $\nabla f(x^k)$ in the norm on the left hand side. Taking expectations, using Assumption 5.3 and the constant $\mu = \max\limits_{x \in \mathcal{X}} \frac{\|x\|_*}{\|x\|_2}$, it holds

$$
\mathbb{E}[\left\| \nabla f(x^k) - g^k \right\|_*] \leq \mu \mathbb{E}[\left\| \nabla f(x^k) - g^k \right\|_2] \leq \mu \sqrt{\mathbb{E}[\left\| \nabla f(x^k) - g^k \right\|_2^2]} \leq \mu\sigma
$$

which we can sum from $k = 1$ to $n$ to obtain

$$
\sum_{k=1}^{n} \mathbb{E}[\left\| \nabla f(x^k) \right\|_*] \leq \frac{\mathbb{E}[f(z^0) - f(z^{n+1})]}{\rho\gamma} + \frac{nL(1-\alpha)\rho\gamma}{\alpha} + 2n\mu\sigma + \frac{nL\rho\gamma}{2}.
$$

Diving both sides by $n$ and then plugging in $\gamma = \frac{1}{\sqrt{n}}$ yields the desired final result. □

## D.4. Linear recursive inequalities

We now present two elementary lemmas that establish bounds for linear recursive inequalities. These results are essential for analyzing the convergence behavior of our stochastic gradient estimator, particularly when examining the error term $\mathbb{E}[\|\lambda^k\|_2^2]$.

**Lemma D.9** (Linear recursive inequality with constant coefficients). *Let $n > 1$ and consider $\{u_k\}_{k=1}^n \in \mathbb{R}_+^n$ a sequence of nonnegative real numbers satisfying, for all $k \in \{2, \ldots, n\}$,*

$$u^k \leq (1 - \beta)u^{k-1} + \eta$$

*with $\eta > 0$ and $\beta \in (0, 1)$. Then, for all $k \in \{2, \ldots, n\}$, it holds*

$$u^k \leq \frac{\eta}{\beta} + (1 - \beta)^k u^1.$$

*Proof.* We prove the claim by induction on $k$. For the base case $k = 2$ we find

$$u^2 \leq (1 - \beta)u^1 + \eta \leq \frac{\eta}{\beta} + (1 - \beta)u^1$$

since $\beta < 1$. Assume now for some $k \in \{2, \ldots, n\}$ that the claim holds. Then, by the assumed recursive inequality on $\{u_i\}_{i=1}^n$, we have

$$u^{k+1} \leq (1 - \beta)u^k + \eta \leq (1 - \beta)\left(\frac{\eta}{\beta} + (1 - \beta)^k u^1\right) + \eta = (1 - \beta)^{k+1}u^1 + \left(\frac{1 - \beta}{\beta} + 1\right)\eta = (1 - \beta)^{k+1}u^1 + \frac{\eta}{\beta}$$

and thus the desired claim holds by induction. $\square$

The first lemma establishes a geometric decay bound for sequences with constant momentum. The following lemma extends this analysis to the case of variable coefficients, which we will use when we analyze Algorithm 1 and Algorithm 2 with vanishing momentum $\alpha_k$.

**Lemma D.10** (Linear recursive inequality with vanishing coefficients). *Let $\{u^k\}_{k \in \mathbb{N}^*}$ be a sequence of nonnegative real numbers satisfying, for all $k \in \mathbb{N}^*$, the following recursive inequality*

$$u^k \leq \left(1 - \frac{1}{2\sqrt{k}}\right)u^{k-1} + \frac{c}{k}$$

*where $c > 0$ is constant. Then, the sequence $\{u^k\}_{k \in \mathbb{N}^*}$ satisfies, for all $k \in \mathbb{N}^*$,*

$$u^k \leq \frac{Q}{\sqrt{k}}$$

*with $Q = \max\{u^1, 4c\}$.*

*Proof.* We prove the claim by induction. For $k = 1$ the inequality holds by the definition of $Q$, since

$$u^1 \leq Q = \frac{Q}{\sqrt{1}}.$$

Let $k > 1$ and assume that

$$u^{k-1} \leq \frac{Q}{\sqrt{k-1}}.$$

Then, by the assumed recursive inequality for $u^k$, we have

$$\begin{aligned}
u^k &\leq \left(1 - \frac{1}{2\sqrt{k}}\right)u^{k-1} + \frac{c}{k} \\
&\leq \left(1 - \frac{1}{2\sqrt{k}}\right)\frac{Q}{\sqrt{k-1}} + \frac{c}{k}.
\end{aligned} \tag{27}$$

Since $k > 1$, we can estimate

$$\frac{1}{\sqrt{k-1}} = \frac{\sqrt{k}}{\sqrt{k(k-1)}} = \frac{1}{\sqrt{k}}\sqrt{\frac{k}{k-1}} = \frac{1}{\sqrt{k}}\sqrt{1 + \frac{1}{k-1}} \leq \frac{1}{\sqrt{k}}\left(1 + \frac{1}{2(k-1)}\right)$$

which, when applied to (27), gives

$$u^k \leq \left(1 - \frac{1}{2\sqrt{k}}\right)\left(1 + \frac{1}{2(k-1)}\right)\frac{Q}{\sqrt{k}} + \frac{c}{k}. \tag{28}$$

Furthermore, as $k > 1$, we also have

$$\left(1 - \frac{1}{2\sqrt{k}}\right)\left(1 + \frac{1}{2(k-1)}\right) \leq \left(1 - \frac{1}{4\sqrt{k}}\right).$$

Applying the above to (28) gives

$$u^k \leq \left(1 - \frac{1}{4\sqrt{k}}\right)\frac{Q}{\sqrt{k}} + \frac{c}{k}$$
$$= \frac{Q}{\sqrt{k}} + \frac{c - Q/4}{k}$$
$$\leq \frac{Q}{\sqrt{k}}$$

with the last inequality following since $Q \geq 4c$. The desired claim is therefore obtained by induction. $\square$

## E. Experiments

### E.1. Additional experiments

**MLP**  We consider a 3-layer MLP with ReLU activations to demonstrate the various output layers in Table 3. We consider the configuration (Spectral → Spectral → X) where X is the output layer. Hyperparameters are provided in Table 9. We observe in Figure 5 that the optimal learning rate transfers across model width for all output layer configurations.

**Shallow GPT**  We consider a 3-layer GPT model (Karpathy, 2023) with the same modernizations as for the deep GPT in Section 6. We additionally remove the weight sharing between the first and last layer so that the various input layers from Table 4 can be investigated. We consider SCION and UNCONSTRAINED SCION with the configuration (X → Spectral → Sign) where X sweeps over the possible input layer lmos. We additionally consider the variant of UNCONSTRAINED SCION using the configuration (Sign → Sign → Sign), which is useful for distributed settings. The hyperparameters can be found in Table 8. We observe in Figure 7 that all configurations exhibit transferability of the optimal stepsize across layer width.

### E.2. Implementation details

It is possible to implement SCG and uSCG, while only storing on set of parameter and one set of gradients (possibly stored in half-precision). For concreteness, we focus on SCG, but the reasoning applies to uSCG as well. Due to the scale invariance of the lmo, uSCG can be equivalently written as

$$G^k = (1 - \alpha)G^{k-1} + \nabla f(x^k, \xi^k)$$
$$x^{k+1} = x^k + \gamma_k \operatorname{lmo}(G^k)$$

By rearranging the update, it suffice to maintain only two states:

$$G \leftarrow G + \nabla f(x, \xi) \quad \text{(backpropagation)}$$
$$x \leftarrow x + \gamma \operatorname{lmo}(G)$$
$$G \leftarrow (1 - \alpha)G$$

Implementation wise this approach relies on storing the averaged gradient at the memory location where backpropagation is accumulating the gradient. Thus, it is important not to zero out the gradient at any point during training. We provide a reference implementation in PyTorch referred to as `ScionLight`.

### E.3. Scaled ReLU$^2$

Large et al. (2024, App. B.2) introduces $\mathrm{ScaledReLU}(x) := \sqrt{2} \cdot \mathrm{ReLU}(x)$ in order to preserve the variance of the input. Building on this we define $\mathrm{ScaledReLU}^2(x) := \mathrm{ScaledReLU}(x)^2 = 2 \cdot \mathrm{ReLU}(x)^2$ as a heuristic.

### E.4. Hyperparameters

For all hyperparameter configuration (Tables 7 to 10) we first tune the radius parameters in (6) on a small proxy model, similar to the input and output scaling factor in $\mu$P (Yang & Hu, 2021). The parameters can be tuned with a suboptimal stepsize $\gamma$. The radius $\rho_1$ refers to the radius of the input layer, the radius $\rho_\ell$ refers to the radius scaling of the intermediary layers, while $\rho_L$ refers to the radius scaling of the last layer in the hyperparameter configuration tables Tables 7 to 11.

All experiments report the loss computed at the last iterate. A linear decay stepsize schedule is employed, which is theoretically motivated by the last iterate guarantee provided in Zamani & Glineur (2023). The linear decay schedule is compatible with any algorithm having regret bounds (Defazio et al., 2024), which was established for SCG in (Hazan & Kale, 2012).

**NanoGPT**   For NanoGPT we specifically build on the version snapshot at:
`https://github.com/KellerJordan/modded-nanogpt/blob/master/records/101724_`
`DistributedMuon/22d24867-eb5a-4fcc-ae2c-263d0277dfd1.txt`.

**ViT**   We train a DeiT-base model on ImageNet using the DeiT codebase (Touvron et al., 2021) in order to reach the reported test accuracy of 81.8%. For the AdamW, we retain the optimized hyperparameters reported in the original code, for both SCION and UNCONSTRAINED SCION, we introduce several revisions.

Specifically, inspired by the NanoGPT architecture, we replace the LayerNorm layers in DeiT-base with RMSNorm (Root Mean Square Normalization), implemented without learnable parameters. Our empirical results indicate that this modification significantly improves the performance of SCION, while it does not produce a similar improvement for AdamW.

Furthermore, we disable the learning rate warmup for SCION and increase the batchsize. The complete set of hyperparameters can be found in Table 11 and the results can be found in Figures 3 and 12.

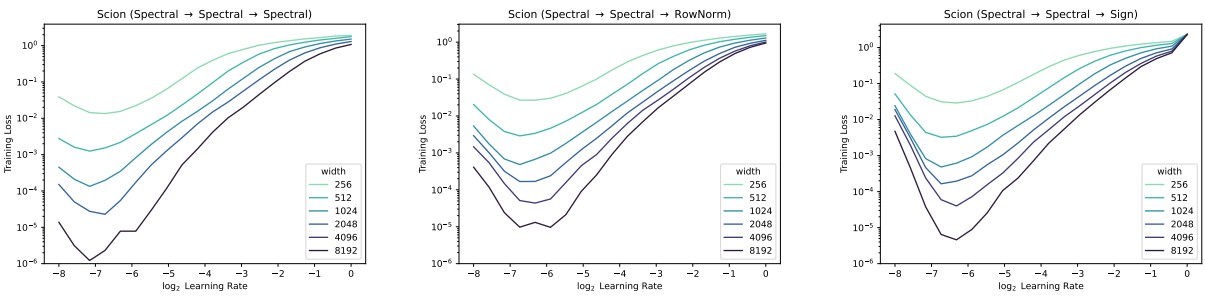

*Figure 5.* Hyperparameter transfer for all three last layer choices on MLP.

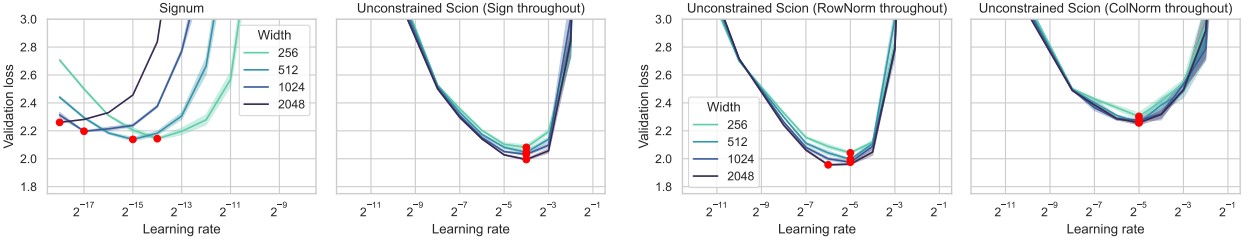

*Figure 6.* Hyperparameter transfer on a 3-layer GPT using appropriately rescaled Sign, RowNorm and ColNorm (*cf.* Table 6).

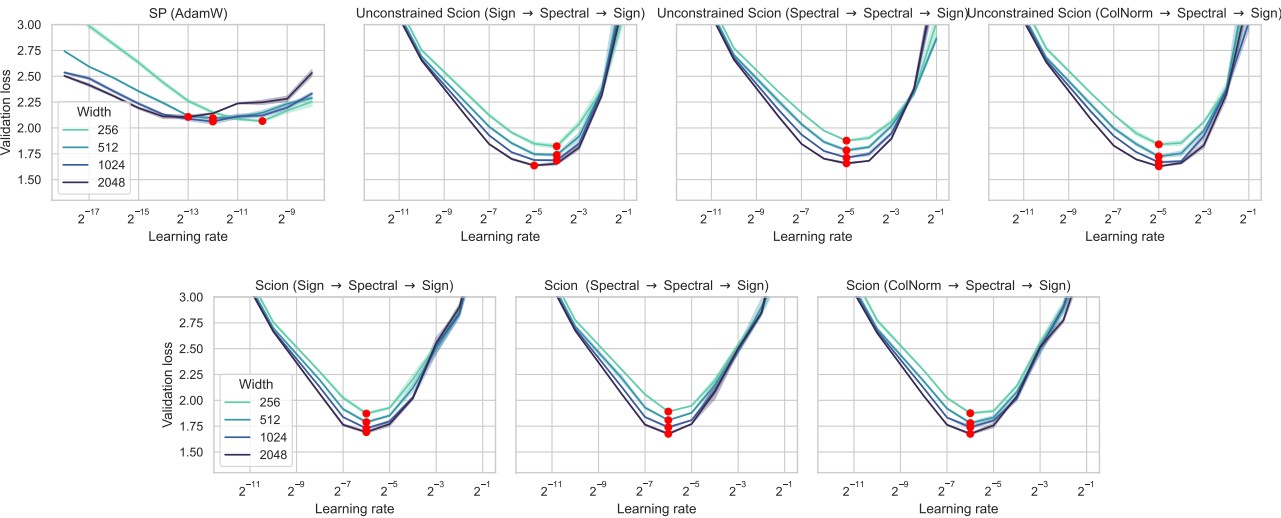

*Figure 7.* Hyperparameter transfer on a 3-layer GPT for all three input layer norms.

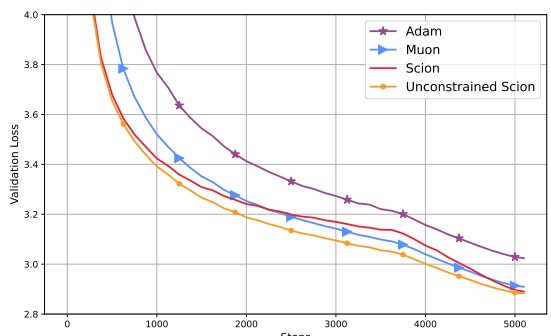

*Figure 8.* Validation loss curve for NanoGPT 3B.

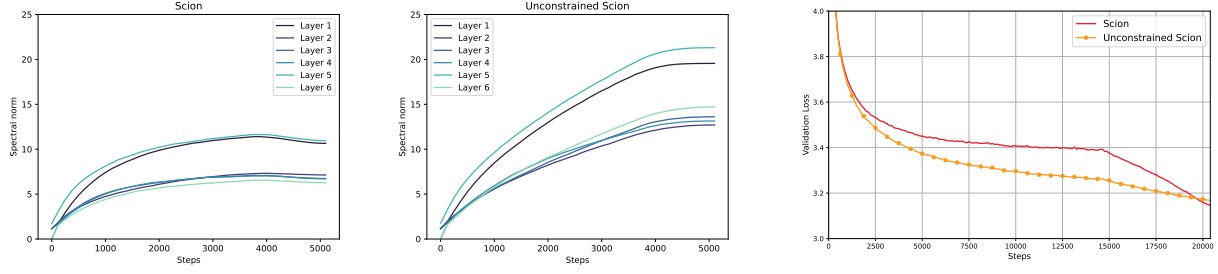

*Figure 9.* (left,middle) Spectral norm of weight matrices on NanoGPT (124M) throughout one training run. Recall that the linear stepsize decay starts at iteration 3650. As expected SCION leads to weights with smaller norms than UNCONSTRAINED SCION. (right) Long run on NanoGPT. The norm control of the constrained algorithm SCION appears to be particularly important for long runs as also observed in Liu et al. (2025) regarding Muon \w weight decay.

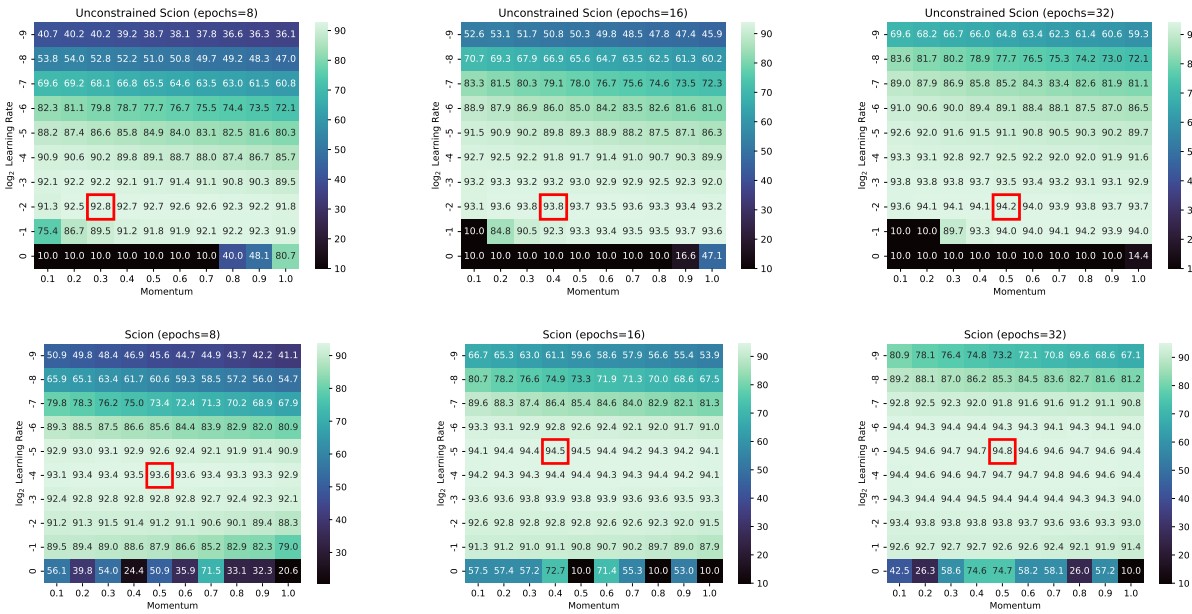

*Figure 10.* The optimal hyperparameters for (UNCONSTRAINED) SCION on the airbench setting with increasing total number of epochs (indicated in red). SCION outperforms UNCONSTRAINED SCION, which is not surprising since norm control is important in the setting.

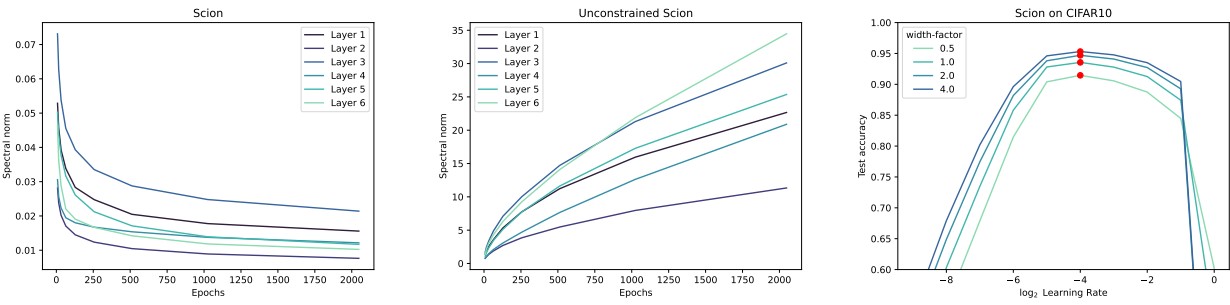

*Figure 11.* (left,middle) Spectral norm of weight matrices on CIFAR10, while sweeping over total number of epochs. The spectral norm grows empirically as $\sqrt{n}$ for UNCONSTRAINED SCION with a fixed stepsize $\gamma$, whereas the norm (provably) stays bounded for SCION. (right) The optimal stepsize transfers across width.

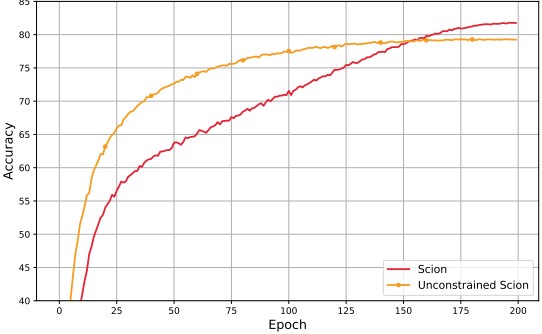

*Figure 12.* Test accuracy curve on the DeiT-base model. The more stringent norm control of SCION is beneficial, similar to what is observed for the CIFAR10 experiments.

Table 7. NanoGPT hyperparameters.

| Hyperparameter | AdamW | Muon | UNCONSTRAINED SCION | SCION |
|---|---|---|---|---|
| Layers | | | 12 | |
| Head dim | | | 128 | |
| Activation function | ReLU$^2$ | | ScaledReLU$^2$ (see Appendix E.3) | |
| Vocabulary size | | | 50304 | |
| Dataset | | | FineWeb | |
| batch size | | | 512 | |
| block size | | | 1024 | |
| Iterations $n$ | | | 5100 | |
| Warmdown | | | 28.5% | |
| Stepsize schedule | | Constant then linear decay $\gamma_k = \begin{cases} \gamma & \text{if } k < n - m \\ \gamma \cdot \left(\frac{n-k}{m}\right) & \text{if } k \geq n - m \end{cases}$ | | |
| Warmup | 5% | | 0 | |
| Gradient clipping | Yes | | No | |
| Momentum $\beta_1$ / $\beta_2$ | 0.9 / 0.95 | | - | |
| Averaging parameter $\alpha$ | - | | 0.1 | |
| Muon stepsize multiplier[1] | - | 0.1 | - | |
| Nesterov | - | Yes | - | |
| Boundary init. | | - | No | |
| Radius $\rho_1$ / $\rho_\ell$ / $\rho_L$ | | - | - / 50 / 3000 | |

[1] Muon uses Adam for the first and last layer. The stepsize for the intermediary layers is multiplied by a constant.

Table 8. Shallow GPT hyperparameters. We increase the batch size to 32, which is the maximum allowed for a model with an embedding size of 4096 on an A100.

| Hyperparameter | AdamW | UNCONSTRAINED SCION | SCION |
|---|---|---|---|
| Layers | | 3 | |
| Head dim | | 64 | |
| Activation function | | $\sqrt{2}\cdot$ GELU (scaled to preserve variance) | |
| Vocabulary size | | 64 | |
| Dataset | | Shakespeare | |
| batch size | | 32 | |
| block size | | 1024 | |
| Iterations $n$ | | 122 | |
| Stepsize schedule | | Linear decay $\gamma_k = \gamma \cdot (1 - k/n)$ | |
| Gradient clipping | Yes | No | |
| $\beta_1$ / $\beta_2$ | 0.9 / 0.95 | - | |
| Averaging parameter $\alpha$ | - | 0.1 | |
| Boundary init. | - | Yes | |
| Radius $\rho_1$ / $\rho_\ell$ / $\rho_L$ | - | 1 / 3 / 10 | |

*Table 9.* Shallow MLP hyperparameters.

| Hyperparameter | SCION |
|---|---|
| Layers | 3 |
| Activation function | ReLU |
| Dataset | CIFAR10 (50000 training examples) |
| batch size | 2048 |
| Epochs | 20 |
| Stepsize schedule | Linear decay $\gamma_k = \gamma \cdot (1 - k/n)$ |
| Averaging parameter $\alpha$ | 0.1 |
| Boundary init. | Yes |
| Radius $\rho_1$ / $\rho_\ell$ / $\rho_L$ | 1 / 1 / 1024 |

*Table 10.* Hyperparameters for the CNN experiments building on the airbench codebase (Jordan, 2024). Batch norm parameters use the Euclidean $\ell_2$ norm and shares scaling factor $\rho_\ell$ with intermediary layers. A further optimized configuration can be found in the associated Github repository, where we also conduct a speedrun matching Muon (with Frobenious normalization) with SCION.

| Hyperparameter | UNCONSTRAINED SCION | SCION |
|---|---|---|
| Block size (block 1, block 2, block 3) | width factor $\times$ (64, 256, 256) | |
| Activation function | GELU | |
| Dataset | CIFAR10 (50000 training examples) | |
| batch size | 2000 | |
| Epochs | 8 | |
| Stepsize schedule | Linear decay $\gamma_k = \gamma \cdot (1 - k/n)$ | |
| Averaging parameter $\alpha$ | 0.5 | |
| Boundary init. | Yes | |
| Radius $\rho_1$ / $\rho_\ell$ / $\rho_L$ | 1 / 1 / 20 | 1 / 1 / 100 |

*Table 11.* DeiT-base hyperparameters. SCION and UNCONSTRAINED SCION uses the configuration (Spectral $\rightarrow$ Spectral $\rightarrow$ Sign) with $\ell_{\mathrm{RMS}}$-norm constraints for the learnable positional embeddings and class tokens, sharing the scaling factor $\rho_\ell$ of the intermediary layers.

| Hyperparameter | AdamW | UNCONSTRAINED SCION | SCION |
|---|---|---|---|
| Layers | | 12 | |
| Head dim | | 64 | |
| Activation function | GELU | $\sqrt{2}\cdot$ GELU (scaled to preserve variance) | |
| Normalization function | LayerNorm | RMSNorm | |
| Sequence Length | | 197 | |
| Dataset | | ImageNet-1k | |
| Stepsize schedule | | Linear warmup and then cosine decay | |
| Max lr | 0.001 | 0.00024 | 0.0004 |
| Warmup epochs | 5 | 0 | |
| Start warmup lr | $10^{-6}$ | - | |
| End lr | $10^{-5}$ | $10^{-7}$ | |
| Batch size | 1024 | 4096 | |
| Epochs | 300 | 200 | |
| $\beta_1$ / $\beta_2$ | 0.9 / 0.999 | - | |
| Averaging parameter $\alpha$ | - | 0.1 | |
| Boundary init. | - | No | |
| Radius $\rho_1$ / $\rho_\ell$ / $\rho_L$ | - | 25 / 25 / 500 | |

