# OpenReview forum: "Training Deep Learning Models with Norm-Constrained LMOs"
_ICML.cc/2025/Conference — ICML 2025 spotlightposter_

### Official Review · Reviewer_2qhG · 2025-03-04

**Overall Recommendation:** 4

**Summary:**

This paper develops an algorithmic framework that can exploit and appropriate choice of norm for entire neural network with emphasis on hyperparameter transfer across model sizes. The algorithm called uSCG, unconstraint stochastic conditional gradient method, shows improvements both theoretically and practically when the norm-ball constraint matches the natural geometry of the problem.

## update after rebuttal
My concerns have been addressed. Good paper.

**Claims And Evidence:**

the evidence presented supports the claims.

**Essential References Not Discussed:**

Not that i am aware of.

**Experimental Designs Or Analyses:**

The experiments are strong enough to support the claims for a mainly theoretical paper.

**Methods And Evaluation Criteria:**

The methods and evaluation both make sense.

**Other Comments Or Suggestions:**

N/A

**Other Strengths And Weaknesses:**

The insight and systemic analysis is quite interesting, unifying everything under lmo gives me a new perspective on looking at things.

**Questions For Authors:**

1. Since you build on top of modded-nanoGPT, I have to ask the question: what is your speedrun result against Muon?

**Relation To Broader Scientific Literature:**

This work is broadly related to series of works that aim to beat AdamW with new optimizer proposal, such as Shampoo, SOAP, Muon. Usually it's under the context of large language model pretraining and finetuning.

**Theoretical Claims:**

The theoretical claims are reasonable though I have not checked the proofs in details.
I really like the part where different optimizers get organized under the lmo framework and presented in a clean and clear way.
Very insightful paper!

---

> ### Author Rebuttal · Authors · 2025-04-01
>
> We are happy that the reviewer found the paper insightful.
>
> > "Since you build on top of modded-nanoGPT, I have to ask the question: what is your speedrun result against Muon?"
>
> The speedrun configuration is the 124M parameter model with 512 batchsize in Fig. 1.
> In this setting Scion consistently across 3 runs has slightly lower validation loss than Muon (with the same wallclock time), which would translate into a slightly better wallclock time if the number of iterations is reduced to exactly match the validation loss of Muon.
>
> The more substantial gain occurs at larger batch sizes.
> We conduct an additional experiment for the batch size 6144 configuration of Fig. 2, where we simply reduce the number of iterations until Scion matches the larger validation loss of Muon.
> We find that Scion can achieve the same validation loss as Muon with a 25% smaller wallclock time in this setting.

---

### Official Review · Reviewer_81Lu · 2025-03-07

**Overall Recommendation:** 2

**Summary:**

The paper introduces the unconstrained Stochastic Conditional Gradient method (uSCG), which builds upon classical Conditional Gradient methods by utilizing linear minimization oracles (LMOs) even for unconstrained optimization problems. This approach leverages the geometry of the problem by assigning different norms for different layers, particularly in language modeling tasks. The authors provide convergence guarantees for uSCG with constant batch sizes and introduce a principled framework for selecting effective constraints based on the input and output space geometries of neural network layers. The paper also discusses the relationship between their method and hyperparameter transfer techniques, such as the Maximal Update Parametrization, which allows for transferring optimal hyperparameters from smaller proxy models to larger ones. The numerical experiments show that the optimal learning rate for the proposed method to minimize validation loss is independent with the model width.

**Claims And Evidence:**

Yes.

**Essential References Not Discussed:**

No.

**Experimental Designs Or Analyses:**

Yes.

**Methods And Evaluation Criteria:**

Yes.

**Other Comments Or Suggestions:**

The algorithmic descriptions in algorithm 1 and 2 are same. Is this a typo?

**Other Strengths And Weaknesses:**

The theoretical novelty of this paper from section 5 is sound and concrete.

Nevertheless, the algorithmic contribution of this paper is unclear. Both algorithm 1 and 2 are simply steepest descent methods with momentum. One minor novelty can be that the authors use different norms for different layers described in section 3.1. Specifically, the authors use Sign in the output layers. According to the numerical experiment, this change does make the optimal learning rate of the proposed method to be independent with the model width but the validation loss improvement is negligible in Figure 1. The experiments on batch size sensitivity does support that the proposed method has better validation loss compares to Muon when batch size is large. Nevertheless, the validation loss curves for all compared algorithms look to be increasing w.r.t. batch size and with the smallest batch size of 512, the difference between proposed method and Muon are not significant.

Besides, the numerical experiments in this paper are restricted to nanoGPT. It would be better to try different transformer architecture to show the applicability of the proposed method in LLM pre-training.

**Questions For Authors:**

No.

**Relation To Broader Scientific Literature:**

Not applicable.

**Theoretical Claims:**

No.

---

> ### Author Rebuttal · Authors · 2025-04-01
>
> We thank the reviewer for their feedback.
> We believe there are important misunderstandings that we address below.
>
> > "The algorithmic descriptions in algorithm 1 and 2 are same. Is this a typo?"
>
> This is not a typo: the two algorithms are distinct and solves different problems as detailed in Section 3.
> Algorithm 1 solves an unconstrained problem, while Algorithm 2 solves a norm-ball constrained problem.
>
> For deep learning this leads to a distinct behaviour of the norm of the weights between Algorithm 1 and 2 as illustrated in Fig. 8-9, since only Algorithm 2 ensures that the norm does not grow beyond the constraint radius $\rho$.
> This explicit norm control can be particularly important for numerical stability and to avoid overfitting, as e.g. seen in the CIFAR10 experiments were Algorithm 2 (Scion) works without the heuristic weight normalization otherwise present in the Muon implementation for this problem.
>
>
> > "Both algorithm 1 and 2 are simply steepest descent methods with momentum"
>
> Neither of our two proposed algorithms is steepest descent (Please see Section 4 were we explicitly compare with steepest descent).
> Algorithm 2 is a Conditional Gradient (aka Frank-Wolfe) based algorithm which applies to constrained problems, while with Algorithm 1 we show that the LMO can even be used for unconstrained problems.
>
> One important distinction with steepest descent is that the stepsize $\gamma$ does not have to depend on the Lipschitz constant $L$ (see our main results Lemma 5.3-5.6).
> Intuitively, this is due to the scale invariance of the LMO, which makes the algorithm more stable (the algorithm is less sensitive to the curvature).
>
>
> > "The experiments on batch size sensitivity does support that the proposed method has better validation loss compares to Muon when batch size is large. Nevertheless, the validation loss curves for all compared algorithms look to be increasing w.r.t. batch size"
>
> The loss is increasing with increasing batch size in the experiments because we keep the total number of tokens fixed.
> The validation loss as a function of batch size is expected to be worse for any optimizer in this setting – the comparison criterion is instead the rate with which performance deteriorates.
> As the reviewer points out, Scion have better validation accuracy than the baselines for larger batch sizes.
>
> One way to quantify the validation loss improvement, is to ask how much quicker Scion can achieve the same validation accuracy as one of the baselines.
> To this end we conduct an additional experiment for the large batch size of 6144, where we simply reduce the number of iterations until Scion matches the larger validation loss of Muon.
> We find that Scion can achieve the same validation loss as Muon with a 25% smaller wallclock time.

---

### Official Review · Reviewer_c2Ed · 2025-03-12

**Overall Recommendation:** 4

**Summary:**

The authors propose a new stochastic family of algorithms exploiting the so-called Linear Minimization Oracle (LMO) to solve optimization problems. It provides a more general framework and unifies many other algorithms as special cases. Theoretical guarantees as well as significant speed-up on the training of models such as nanoGPT demonstrate the effectiveness of their proposal.

**Claims And Evidence:**

All the claims and experimental evidence are clearly presented and justified. I hope the code is publicly available once the paper is published.

**Essential References Not Discussed:**

One particular line of related research (that is quite old) is natural gradient descent [1]. It also exploits the geometry of the function by defining a pseudo-metric in the parameters space by KL-divergence. It is not very popular nowadays since: 1) Calculating the Fischer matrix and its inverse is quite expensive, and 2) The empirical performance in stochastic setting is not very impressive. However, I think it is worth mentioning because it is among the first papers stepping away from the Euclidean metric and proposing a new metric for optimization, to the best of my knowledge.

[1]: S. Amari, "Natural Gradient Works Efficiently in Learning," in Neural Computation, vol. 10, no. 2, pp. 251-276, 15 Feb. 1998, doi: 10.1162/089976698300017746.

**Experimental Designs Or Analyses:**

I think the experimental designs are justified and credible.

**Methods And Evaluation Criteria:**

The proposed framework is pretty interesting and well-presented. The experiments conducted look credible and show certain improvement over existing methods.

**Other Comments Or Suggestions:**

N/A

**Other Strengths And Weaknesses:**

N/A

**Questions For Authors:**

I have one question: From what I understand, all the LMOs calculated in Tables 2, 3 and 4 do not take into account the non-linear activation function. If that is the case, do you think taking them into account can improve the optimization process? After all, the geometry of the data after the activation is what really matters, and not the pre-activated ones.

**Relation To Broader Scientific Literature:**

The paper provides a general framework to unify several methods in the literature.

**Theoretical Claims:**

I found all theoretical claims sound. I admit that I did not check the proof very carefully in the Appendix. However, for the parts that I did check, there is not any mathematical error.

To be verified: In Lemma D.1, I think it is the right constant is $L_2$, and not $L$.

---

> ### Author Rebuttal · Authors · 2025-04-01
>
> We thank the reviewer for their positive feedback.
>
> > "To be verified: In Lemma D.1, I think it is the right constant is $L_2$, and not $L$."
>
> The right constant is $L$, since smoothness is taken w.r.t. the arbitrary norm $\Vert\cdot\Vert$.
> This is what eventually yields the dependency on $L$ seen in the main results of Section 5.
>
> What might have caused the confusion is that there is a minor typo in the statement of Lemma D.1, since the guarantee of Lemma D.1 should be stated in terms of $\Vert\nabla f(\bar x^n)\Vert_*$ instead of $\Vert\nabla f(\bar x^n)\Vert_2^2$.
> This is just a typo though, that does not change the final result, since the following Lemma 5.3 is correctly stated in terms of the dual norm.
> We will correct the typo in the final version.
>
> > "One particular line of related research (that is quite old) is natural gradient descent [1]. It also exploits the geometry of the function by defining a pseudo-metric in the parameters space by KL-divergence. It is not very popular nowadays since: 1) Calculating the Fischer matrix and its inverse is quite expensive, and 2) The empirical performance in stochastic setting is not very impressive. However, I think it is worth mentioning because it is among the first papers stepping away from the Euclidean metric and proposing a new metric for optimization, to the best of my knowledge. "
>
> This is a good point. We will remember to cite and discuss natural gradient descent.
>
> > Geometry of activation functions
>
> It might be possible to further improve the performance by taking the geometry of activation functions into account, but it seems challenging.
> One place to maybe look for ideas is the vast literature on initialization.

---

> > ### Comment · Reviewer_c2Ed · 2025-04-02
> >
> > Thanks the authors for their responses. I have no further questions and I keep the score as it is.

---

### Official Review · Reviewer_WPwo · 2025-03-14

**Overall Recommendation:** 3

**Summary:**

This paper proposes a new approach to optimizing DNNs that leverages the Linear Minimization Oracle (LMO) over norm-constrained sets. The core idea is to adapt the optimizer a priori to the geometry of the neural network, rather than relying solely on on-the-fly adaptive methods like Adam. The authors introduce a new stochastic algorithm, unconstrained Stochastic Conditional Gradient (uSCG), which surprisingly applies the LMO concept to unconstrained problems. They also adapt the existing constrained Stochastic Conditional Gradient (SCG) method for non-convex deep learning. The key to their approach is choosing a norm for the LMO that reflects the structure of the neural network, particularly operator norms between layers. The paper claims that this approach leads to improved performance, better batch size robustness, memory efficiency, and zero-shot hyperparameter transferability across models of different widths. The authors provide theoretical convergence guarantees and empirical results on nanoGPT language modeling and CIFAR-10 image classification tasks. The framework unifies several known optimization algorithms like Normalized SGD, SignSGD, and Muon.

**Claims And Evidence:**

The claims are generally supported by evidence. However, while the claim of improved performance and hyperparameter transferability is well supported for language modeling, the evidence is weaker for image classification tasks, lacking comparison to other optimizers in this domain.

**Essential References Not Discussed:**

None.

**Experimental Designs Or Analyses:**

The nanoGPT experiments are well-designed to test the core claims on language modeling. Varying model width, batch size, and comparing to Adam and Muon provides a strong evaluation. The CIFAR-10 experiments show the transferability of the optimal stepsize, but they do not establish the method's competitiveness in image classification.

**Methods And Evaluation Criteria:**

The core methodological idea of using norm-constrained LMOs, and particularly the uSCG algorithm, is novel and makes sense for adapting the optimizer to the network structure. The connection to operator norms is theoretically well-motivated. The nanoGPT experiments, including the use of a 3B parameter model, are appropriate for evaluating performance and hyperparameter transfer in language modeling. The CIFAR-10 experiments are insufficient for evaluating the method's general applicability. Comparison to other optimizers should be made on more diverse datasets.

**Other Comments Or Suggestions:**

The paper could benefit from a clearer explanation of the intuition behind why uSCG works for unconstrained problems.

**Other Strengths And Weaknesses:**

None.

**Questions For Authors:**

Comparison to other optimizers should be made on more diverse datasets

**Relation To Broader Scientific Literature:**

The paper connects to several areas of the literature, including conditional gradient methods and adaptive optimization.

**Theoretical Claims:**

I didn't thoroughly check the correctness of the proofs.

---

> ### Author Rebuttal · Authors · 2025-04-01
>
> We thank the reviewer for their feedback and address the remaining concerns below.
>
> > "The CIFAR-10 experiments show the transferability of the optimal stepsize, but they do not establish the method's competitiveness in image classification"
>
> To establish competitiveness on image classification we conduct a speedrun to achieve the 94% test accuracy target on the [airbench](https://github.com/KellerJordan/cifar10-airbench) codebase.
> Scion achieves a 94.08% test accuracy (averaged over 200 runs) in the same wallclock time that the SOTA method Muon uses to achieve the target accuracy.
> This provides a 13.6% speedup over the tuned baseline of SGD with momentum.
>
> Note that Scion achieves this SOTA performance without the ad-hoc weight normalization otherwise used in the airbench implementation of Muon.
>
> > "The paper could benefit from a clearer explanation of the intuition behind why uSCG works for unconstrained problems."
>
> It was indeed also surprising to us initially that we could show convergence for uSCG.
>
> After applying smoothness in the analysis there are mainly two terms to address: i) the _direction_ of the update (given by an inner product), and ii) the _magnitude_ of the update.
> For simplicity, let us consider the deterministic case where we have
>
> $f(x^{k+1}) \leq f(x^k) + \gamma\rho \langle \nabla f(x^k), \operatorname{lmo}(\nabla f(x^k)\rangle + \tfrac{\gamma^2\rho^2 L}{2}$
>
> The main insight is the relationship between the LMO over a norm-ball and the definition of the dual norm. Specifically, the value that the LMO attains on the boundary of the norm-ball is exactly the dual norm of the gradient. So the _direction_ of the update (the inner product) is still informative and it in fact decreases the function value by the dual norm of the gradient.
> However, the _magnitude_ of the update (the last term) will be large even close to a solution, but this can be taken care of by the stepsize choice (as made precise by Lemma 5.3 and 5.4).
>
> Notice that we do not need to know the Lipschitz constant $L$ to select the stepsize $\gamma$ as is otherwise the case in e.g. gradient descent. The intuitive reason for this is the scale invariance of the LMO.
> We will remark on these intuitions in the final version.

---

### Decision · Program_Chairs · 2025-05-01

**Decision:**

Accept (spotlight poster)

**Comment:**

This paper proposes a novel stochastic optimization framework, Stochastic Conditional Gradient (SCG) and unconstrained SCG (uSCG), which leverages the Linear Minimization Oracle (LMO) to adapt to problem geometry in deep neural networks. The framework unifies existing methods like Normalized SGD, SignSGD, and Muon. By instantiating SCG/uSCG with operator norms (dubbed Scion/unconstrained Scion), the method achieves hyperparameter transferability across model sizes and competitive performance on tasks such as nanoGPT and CIFAR-10.

While many optimizers have claimed to beat Adam in recent years, but none of them actually did. Recent consensus suggests Muon  may achieve this.    This paper theoretically and empirically demonstrate that Scion/unconstrained Scion outperform Muon and Adam. Moreove, Scion/unconstrained Scion have another appealing character — zero-shot hyperparameter transferability across models of different widths,  which means small proxy models can tune hyperparameters for large models, significantly boosting LLM training efficiency. This positions Scion as a potential default optimizer for DNNs, akin to Adam’s current dominance.

Three reviewers strongly endorse the paper; one raised concerns, but the authors addressed these in the rebuttal. Given the method’s technical novelty, empirical success, and broad applicability, the AC strongly recommends acceptance.